# Direct measurement of the Criegee intermediate CH$_2$OO in ozonolysis of ethene

Mixtli Campos-Pineda [1,3,4], Lei Yang [1,4] & Jingsong Zhang [1,2] ✉

The transient species produced from reactions of unsaturated hydrocarbons with ozone, carbonyl oxides, termed "Criegee intermediates", play a key role in tropospheric oxidation mechanisms. Direct observation and characterization of Criegee intermediates in ozonolysis in situ were proven difficult in decades of efforts. Here, we report the direct measurement of the simplest Criegee intermediate, CH$_2$OO, from ozonolysis of ethene by cavity ring-down spectroscopy in a flow cell reactor. The transient CH$_2$OO is quantified rapidly by near-ultraviolet absorption spectra via its $\tilde{B}(^1A') \leftarrow \tilde{X}(^1A')$ transition. Time profiles of CH$_2$OO produced in ozonolysis under quasi-steady state conditions are observed. These CH$_2$OO concentration profiles benchmark the modeling of the ethene ozonolysis reaction network and mechanism, allowing for determination of the yield and various kinetic data of CH$_2$OO.

Ozonolysis of olefins plays an important role in the troposphere as it is one of the main oxidation processes of unsaturated volatile organic carbons[1–3]. Products of ozonolysis can ultimately lead to the production of highly oxidized molecules, secondary organic aerosols, carbonyl products, organic radicals, and hydroxyl radical[4–7]. The oxidizing capacity of this reaction lies in its mechanism, first described in the liquid phase by Rudolph Criegee[8], which involves a 1,3 dipolar cycloaddition of ozone to the olefinic bond, through a van der Waals complex, leading to the formation of a primary ozonide (POZ)[9,10]. This reaction is highly exothermic and the POZ breaks rapidly into a carbonyl and a carbonyl oxide product with broad internal energy distributions. The carbonyl oxide product, also known as Criegee intermediate (CI), is a transient species that is involved in the oxidation processes mentioned above either via isomerization, decomposition, or bimolecular reactions. Carbonyl oxides with low internal energy will have a long enough lifetime to undergo bimolecular reactions and are called "stabilized" Criegee intermediates (sCIs)[11–14], whereas highly energetic carbonyl oxides are known as "hot" CIs.

For decades, Criegee intermediates have been the subject of extensive studies. The breakthrough work of Welz et al.[15] devised a method for the synthesis of sCIs by photolysis of diiodoalkanes and subsequent reaction of the iodoalkyl radicals with oxygen. This method has been exploited by many groups to measure ultraviolet[16–18] and infrared[19] spectra, study fundamental unimolecular processes[20,21],

and determine kinetic rate constants for several bimolecular reactions of atmospheric interest[15,22–25]. The first of the carbonyl oxides studied in this manner was formaldehyde oxide, CH$_2$OO[15], leading to extensive kinetic information on its bimolecular reactions[15,22,23], as well as its unimolecular decomposition and isomerization processes[16,24]. However, due to the highly exothermic POZ decomposition following 1,3 dipolar cycloaddition of ozone to the olefinic bond, the production of Criegee intermediates in ozonolysis is accompanied by high internal energy[9,26,27]. Experimental studies of ethene ozonolysis and theoretical computations of the energetics of formaldehyde oxide produced from the scission of the POZ show that the nascent yield of "stabilized" CH$_2$OO is 0.20 ($\pm$0.003)[11,12,28], while it increases to 0.42 ($\pm$0.1) at atmospheric pressure due to collisional stabilization[29]. In addition, its low OH yield (0.17 $\pm$ 0.05)[29,30] is strong evidence that the absence of $\alpha$-hydrogens leads to isomerization of the "hot" CI into dioxirane, and its subsequent processes lead to fragments that participate in further reactions. Hence, both the "hot" and "stabilized" fractions of CH$_2$OO add to the complexity of the ozonolysis mechanism, and more information on this branching and the secondary processes becomes important for a better understanding of the ozonolysis reaction in the troposphere[31]. Therefore, it is necessary to study the ozonolysis reaction itself by monitoring the important species and comparing experimental measurements with a model mechanism that contains accurate pathways and rate constants. CIs, as the immediate fragment

[1]Department of Chemistry, University of California, Riverside, CA, USA. [2]Air Pollution Research Center, University of California, Riverside, CA, USA. [3]Present address: Centre for Research into Atmospheric Chemistry, University College, Cork, Ireland. [4]These authors contributed equally: Mixtli Campos-Pineda, Lei Yang. ✉e-mail: jingsong.zhang@ucr.edu

of POZ that determine the reaction branching and connect to subsequent reaction pathways in ozonolysis, control the outcomes of the ozonolysis reactions and are thus the most crucial transient species in ozonolysis. Direct observation and kinetic measurements of CIs in ozonolysis in situ in real time will anchor the reaction mechanisms and greatly improve our fundamental understanding of the whole reaction network.

Decades of efforts have proven the difficulty in measuring Criegee intermediates directly from gas-phase ozonolysis in situ due to their transient nature, high reactivity, and low concentrations (from slow production and fast reactions and decomposition)[32,33]. Recently, Womack et al.[33] reported the first direct observation of $CH_2OO$ in ozonolysis of ethene using the subtle signals attributed to $CH_2OO$ measured by Fourier transform microwave spectroscopy and pulsed nozzle over a very long time signal integration (4.3 h or 93,000 sample injections) but at only one poorly-defined residence time (estimated to be <0.5 s with a 6-Hz sampling rate). In this work, we couple a flow cell reactor with cavity ring-down spectroscopy (CRDS) and exploit the high sensitivity of the multi-pass absorption spectroscopy technique to directly observe and measure kinetics of the transient $CH_2OO$ intermediate in ozonolysis of ethene at short residence times. The measured near-ultraviolet (near-UV) spectra are compared to a literature reference spectrum of $CH_2OO$, allowing for determination of the number densities. The strong absorption features of $CH_2OO$ with good signal-to-noise ratios facilitate direct kinetic and mechanistic studies with time profiles of $CH_2OO$ in actual ozonolysis systems. Time-dependent concentrations of HCHO product and consumed $O_3$ are also determined similarly in separate measurements under the same reaction conditions. A mechanism of ethene ozonolysis is constructed with available kinetic data and constrained with the measured time-dependent number densities to assess the best-fit kinetic rate constants of reactions that are important secondary steps in the ozonolysis of alkenes. The detailed experimental and kinetic simulation methods are described in Method Section and supplementary materials.

## Results

### Detection of $CH_2OO$

Figure 1 shows the high-resolution absorption spectrum of $CH_2OO$ in situ from ethene ozonolysis, where the absorption cross-sections were scaled from measured absorption coefficient in comparison with $CH_2OO$ reference spectra by Foreman et al.[34]. Under the reaction condition of this spectrum in the region of 363-395 nm (initial ozone concentration of $1.8 \times 10^{15} cm^{-3}$, initial ethene concentration of $1 \times 10^{17} cm^{-3}$, residence time of ~140 ms, and pressure of 10 Torr), all the absorptions except that from $CH_2OO$ were considered negligible, and the number density of $CH_2OO$ was determined to be $2.75 \times 10^{11} cm^{-3}$. Note that the residence time in the plug flow reactor here represents reaction times spanning from 0 to the nominal total residence time (e.g., the 140 ms residence time represents reaction times from 0 to 140 ms), with the measurement integrating the signal from the point of injection up to the nominal total residence time, probing an average across a horizontal slice in Supplementary Fig. 1. The spectral resolution in this work (0.01 nm) is higher compared to those in the references (~0.12–2 nm)[16–18,34,35]. Vibronic bands of the $\tilde{B}(^1A') \leftarrow \tilde{X}(^1A')$ transition of $CH_2OO$, which originate from excitation to the bound levels of the $\tilde{B}$ state rather than from hot bands[34], show excellent agreement with the reference spectra[17,34,35]. The good signal-to-noise ratio in this work allowed the determination of $CH_2OO$ concentration using the vibronic band features spaced by ~8 nm (or 600 cm$^{-1}$) with half-peak widths of ~3.5 nm (or 200 cm$^{-1}$) in the following kinetic experiments. The relative uncertainties in $CH_2OO$ absorption cross sections in the references listed (15–30%[17,34,35]) are larger than our spectra uncertainties (1σ error bar estimated to be ~3–10% from repeated measurements, see Source Data for Fig. 3), therefore the

uncertainties of the cross sections determined in this work by scaling to the reference cross sections from Foreman et al. are ~30%[34] and could be improved when more accurate $CH_2OO$ reference become available in the future. HCHO was not produced in high enough concentration in the short residence times to affect the absorption spectra of $CH_2OO$ in 363–395 nm. Only when residence time and reactant concentrations were increased by more than 10 times, weak rovibronic features from the $\tilde{a}^3A_2 \leftarrow \tilde{X}^1A_1$ transition of HCHO[36,37] can be identified in this wavelength region (see Supplementary Fig. 2). This confirms that residence times smaller than 500 ms were short enough to avoid spectra interference from other reaction byproducts along the reaction cell.

Further confirmation that the absorption features observed belong to $CH_2OO$ was obtained by adding sulfur dioxide to the ozonolysis reaction via the $C_2H_4 + N_2$ flow to scavenge $CH_2OO$ with a rate constant of 3.7 ($\pm 0.5$) $\times 10^{-11} cm^3 s^{-1}$ [(29)]. The concentration of $SO_2$ used was $9 \times 10^{14} cm^{-3}$, more than a thousandfold the concentration of $CH_2OO$. Figure 2 shows the change in absorption coefficient for the ozonolysis reaction with and without $SO_2$. With the addition of $SO_2$, the features attributed to $CH_2OO$ vanish and only the spectra of $SO_2$ remain. This chemical titration is a strong indication that the absorption features observed indeed belong to $CH_2OO$. Supplementary Fig. 3 further shows that the $CH_2OO$ concentration decreased by more than 95% after the excess amount of $SO_2$ was added.

Ethene and ozone were introduced to the flow reactor in excess concentrations, becoming the main drivers of the chemical processes. Therefore, the rate constants of their reactions with $CH_2OO$ can be measured with some degree of precision. Residence time gradually increased from 9 to 434 ms and the average concentrations of $CH_2OO$ in the flow reactor were measured. Figure 3 illustrates the concentrations of $CH_2OO$ from experiments conducted at various residence times and under different ethene and ozone concentrations (depicted by solid symbols) at 10 Torr, showing clear production and depletion time profiles that are modeled and compared to kinetic simulation (represented by open symbols; discussed next). The time profiles (9–434 ms residence time) of Criegee intermediate produced in

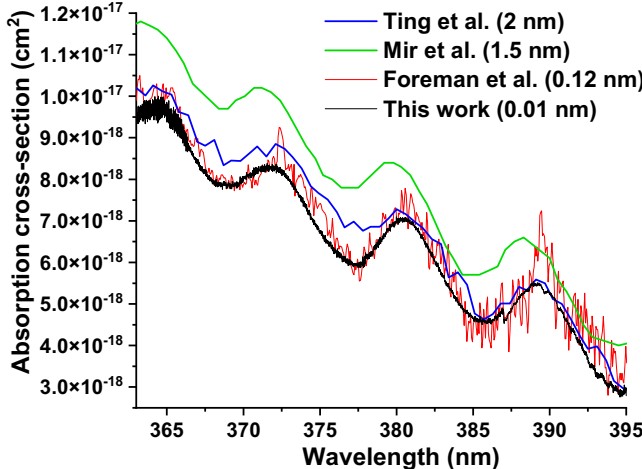

**Fig. 1 | Absorption cross-sections of the 363–395 nm section of the $\tilde{B}(^1A') \leftarrow \tilde{X}(^1A')$ transition of $CH_2OO$ produced in ethene ozonolysis.** The reaction conditions were initial ozone concentration of $1.8 \times 10^{15} cm^{-3}$, initial ethene concentration of $1 \times 10^{17} cm^{-3}$, and residence time of ~140 ms. The laser scanning step was 0.01 nm. The cross-sections in this work (in black) were determined by scaling the absorption coefficient to the reference cross-section data from Foreman et al.[34] (in red), and compared to Ting et al.[17] (in blue) and Mir et al.[35] (in green), adapted with permission from American Chemical Society and Royal Society Chemistry. The resolution of the reference spectra are labeled. Source data are provided as a Source Data file.

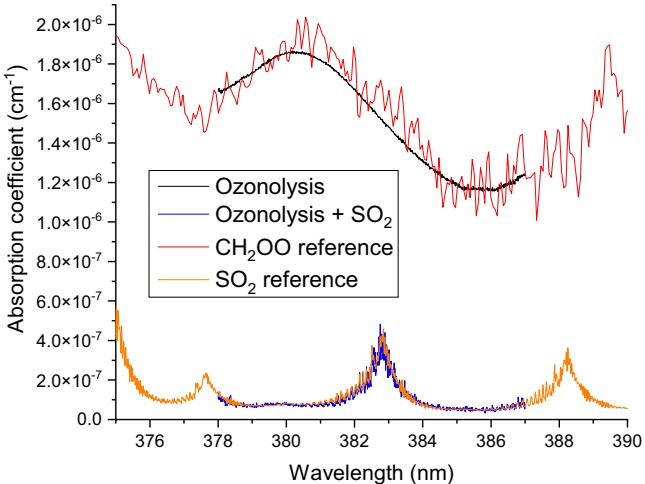

**Fig. 2 | Absorption spectra measured during ozonolysis of ethene with the presence (in blue) and absence (in black) of high-concentration SO$_2$ (used as a scavenger).** The CH$_2$OO concentration decreased by more than 95% after the addition of the scavenger (see enlarged Supplementary Fig. 3). The SO$_2$ reference spectra (Vandaele et al.[77], in orange) were obtained from the MPI-Mainz UV/VIS Spectral Atlas[62]. The CH$_2$OO reference (in red) was scaled from cross-section data of Foreman et al.[34] Source data are provided as a Source Data file.

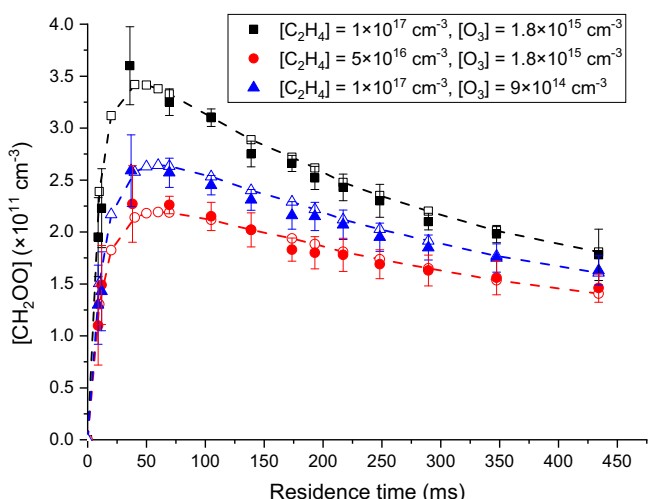

**Fig. 3 | Concentration time profile of CH$_2$OO in ethene ozonolysis at different flow rates (residence times) under different reaction conditions at 10 Torr and 293 K.** (solid symbols: experimental data; open symbols: kinetic simulation; black squares: initial ethene concentration [C$_2$H$_4$] = 1×10$^{17}$ cm$^{-3}$ and initial ozone concentration [O$_3$] = 1.8×10$^{15}$ cm$^{-3}$; red circles: [C$_2$H$_4$] = 5×10$^{16}$ cm$^{-3}$ and [O$_3$] = 1.8×10$^{15}$ cm$^{-3}$; blue triangles: [C$_2$H$_4$] = 1×10$^{17}$ cm$^{-3}$ and [O$_3$] = 9×10$^{14}$ cm$^{-3}$). The dashed lines connect the kinetic simulation open symbols under the different reaction conditions. The concentration of CH$_2$OO at 174 ms with [C$_2$H$_4$] = 1×10$^{17}$ cm$^{-3}$ and [O$_3$] = 1.8×10$^{15}$ cm$^{-3}$ was measured three times for the 1σ error bar, while spectra baseline fluctuations were used to estimate the error bars of other data points. Source data are provided as a Source Data file.

ozonolysis were observed. Notably, the CH$_2$OO concentration quickly ascends from zero upon the mixing of ethene and ozone. Subsequently, the interplay of bimolecular reactions and (to a lesser extent) unimolecular decomposition of sCIs becomes evident in consuming CH$_2$OO, making the net production of CH$_2$OO slow down and reach its maximum concentration around 40–60 ms. Beyond the inflection point, the consumption rates of CH$_2$OO surpass its production, leading to a steady decline in the concentration of CH$_2$OO over the longer

residence times. The concentration of CH$_2$OO at 174 ms with relatively high alkene and high ozone concentrations (solid black square) was measured three times to generate the 1σ error bar (8×10$^9$ cm$^{-3}$) at that point, while spectra baseline fluctuations (Δα) were used to estimate the error bars of other data points (-1–3×10$^{10}$ cm$^{-3}$) in Fig. 3 (assuming no additional uncertainties from the CH$_2$OO reference). The changes in absorption spectra of the three kinetic curves are shown in Supplementary Fig. 4.

### Reaction network simulation and reaction mechanism

Prior to the experiments, a preliminary kinetic simulation was first built and utilized to optimize the initial experimental conditions for CH$_2$OO production at different residence times. It was further revised later after comparison with the experimental measurements. The plug flow reactor model was validated with parameters listed in Supplementary Table 1 and simulated as a series of continuous stirred tank reactors (CSTR), where the output of a CSTR becomes the input of the next one along the flow cell. A full description of the revised kinetic model and ethene ozonolysis mechanism can be found in Supplementary Table 2, and a simplified schematic mechanism is in Fig. 4. Supplementary Table 3 summarizes and compares the pseudo-first-order reaction rates of CH$_2$OO consumption reactions at 10 ms and 434 ms residence time. At the beginning stage of 10 ms, the reactions with O$_3$ and ethene are the major consumption pathways (~56% and ~14%) of CH$_2$OO. The bimolecular self-reaction of CH$_2$OO also plays an important role (~25%) due to its fast reaction rate constant (7.40 × 10$^{-11}$ cm$^3$ s$^{-1}$). As the reaction goes on, byproducts produced in ozonolysis and secondary reactions accumulate and gradually become important consumption pathways of CH$_2$OO. At 434 ms, the reactions with HCHO, HCOOH (from CH$_2$OO + HCHO), and O$_3$ are the major reactions of CH$_2$OO, taking up to ~45%, ~21%, and ~17% of its consumption, respectively. From 10 ms to 434 ms, the total pseudo-first-order consumption rate of CH$_2$OO increases from 139 s$^{-1}$ to 452 s$^{-1}$, illustrating that it is important to keep residence time short to maximize the concentration of CH$_2$OO. The plot of percentage contributions to CH$_2$OO loss as a function of residence time is presented in Supplementary Fig. 5. Supplementary Fig. 1 depicts the concentration profile of CH$_2$OO along the reactor at various residence times within the 500 ms timeframe, obtained from kinetic simulation conducted with the initial ethene and ozone concentrations of 1 × 10$^{17}$ and 1.8 × 10$^{15}$ cm$^{-3}$, respectively. This contour plot also underscores the importance of maintaining a short residence time to guarantee a sufficiently detectable concentration of CH$_2$OO.

Rate constants of most of the bimolecular reactions of CH$_2$OO that play a role in the ozonolysis mechanism were obtained from kinetic studies of CH$_2$OO found in the literature, as explained above. An exception is the rate constant of the CH$_2$OO reaction with C$_2$H$_4$ (k = 7 (±1) ×10$^{-16}$ cm$^3$ s$^{-1}$) measured by Buras et al.[25] In their work they suggested this rate constant to be considered as a lower limit. Theoretical calculation by Sun et al.[38] found the rate constant to be higher (3.91 × 10$^{-15}$ cm$^3$ s$^{-1}$). However, we found the best fit of the model when the rate constant of CH$_2$OO + C$_2$H$_4$ was set to 2 (±0.2) × 10$^{-16}$ cm$^3$ s$^{-1}$ in order to account for the difference in the average concentration of CH$_2$OO when the total rate of production is kept the same but the ratio of C$_2$H$_4$ and O$_3$ changes. Error analysis was performed by randomly varying corresponding rate constants 100 times assuming a Gaussian distribution to generate the 1σ standard deviations, as shown with the error bars in kinetic simulation (colored shades) in Supplementary Fig. 6. The upper limits of the error bars of kinetic data were confirmed when the modeling results apparently mismatched with the experimental trends. The experimental data also indicate that the rate constant of the reaction of CH$_2$OO and O$_3$ must be higher than that for CH$_2$OO and C$_2$H$_4$ as more CH$_2$OO remains when the initial concentration of ozone decreases, indicating that ozone contributes more to CH$_2$OO depletion. The

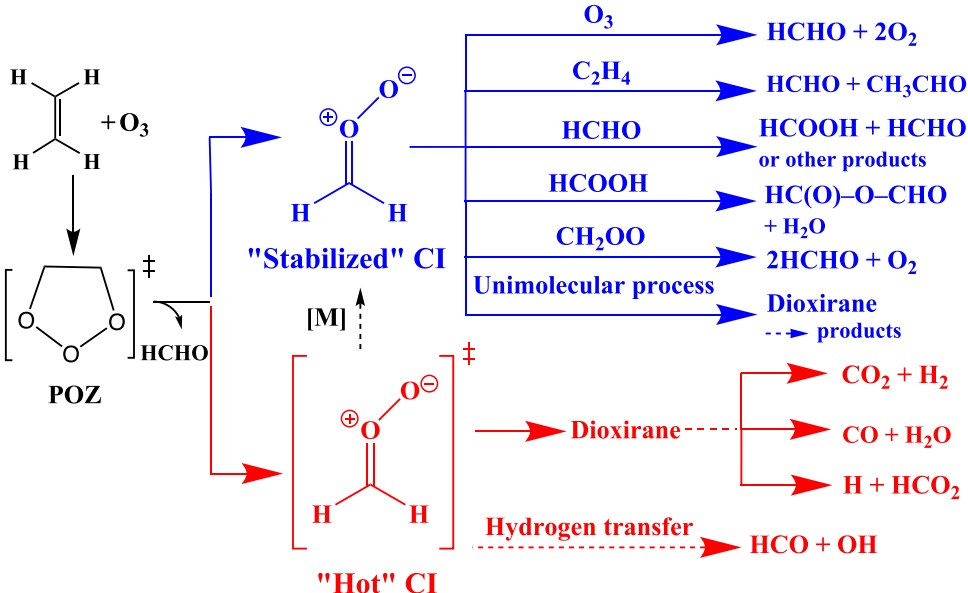

**Fig. 4 | Simplified reaction network of ethene ozonolysis under laboratory condition, emphasizing primary reaction channels of "stabilized" and "hot" CH₂OO.** Detailed reaction mechanism is in Supplementary Table 2.

best fit to experimental data occurs when the rate constant of the reaction of $CH_2OO$ with $O_3$ is set to 4.5 ($\pm$0.5) $\times10^{-14}$ cm³ s⁻¹. Theoretical work from Vereecken et al.[39,40] obtained a rate constant of $4\times10^{-13}$ cm³ s⁻¹. Experimental work by Onel et al.[41] reported this rate constant to be 3.6 ($\pm$0.8) $\times10^{-13}$ cm³ s⁻¹. However, in our ethene ozonolysis mechanism, it was found that a rate constant higher than $2\times10^{-13}$ cm³ s⁻¹ would completely inhibit the nonlinear behavior in the depletion of $CH_2OO$ at longer residence times, as ozone would become the only important depletion process. Copeland et al.[42] find their data to be best fitted with a rate constant of $CH_2OO+O_3$ of $1\times10^{-13}$ cm³ s⁻¹. Chang et al.[43] reported a rate constant of 6.7 ($\pm$0.8) $\times10^{-14}$ cm³ s⁻¹. Our best-fit rate is more consistent with that by Chang et al.[43].

As a main product of ethene ozonolysis, formaldehyde also drives secondary chemical processes and is likely to account for the nonlinear depletion of $CH_2OO$ in our experiments. Copeland et al.[42] used product branching ratios of the $CH_2OO+HCHO$ to obtain a best-fit total rate constant of $9.2\times10^{-13}$ cm³ s⁻¹. However, our best fit occurs when the total rate constant is set to 3.1 ($\pm$0.3) $\times10^{-12}$ cm³ s⁻¹. Recently, Luo et al.[44] reported this rate constant to be 4.11 ($\pm$0.25) $\times10^{-12}$ cm³ s⁻¹, showing consistency with our result. In addition, our model shows only slight differences in the average concentrations of $CH_2OO$ when different branching ratios are used for the different pathways of $CH_2OO$ with formaldehyde, preventing any meaningful assignment of its branching ratios in our model. More work is, therefore, needed to assess the depletion of $CH_2OO$ by HCHO, and the importance of its reaction products in subsequent reaction pathways.

From the fitting of kinetic modeling with the experimental [$CH_2OO$] data, the yield of stabilized $CH_2OO$ was determined to be 0.25 ($\pm$0.07) in the low-pressure region (~10 Torr). The error bar of the sCI yield from kinetic simulation (relative error ~4%, see Supplementary Fig. 6) was much smaller than the uncertainty of [$CH_2OO$] originated from the $CH_2OO$ reference spectra (relative error ~30%), with the latter contributing to most of the sCI yield uncertainty. Previously, the yield of sCIs in ethene ozonolysis was measured to be ~0.19–0.25 below 20 Torr using chemical titration reactions of sCIs with excess scavenger $SO_2$[11,28]. Theoretical studies using statistical models reported the nascent yield of 0.2 at the zero-pressure limit[12], while more recent calculations using both statistical

and trajectory models reported 0.36[9,26]. Our result agrees well with the yields from the chemical titration method and the statistical models. When the pressure changed from 4 to 19 Torr, the stabilized $CH_2OO$ yield showed a modest increase (0.23–0.25) ($\pm$0.07), consistent with previous studies as shown in Supplementary Fig. 7.

The consumption of ozone and production of HCHO were measured in the 325–340 nm range under the same reaction conditions to measure the concentration and yield of formaldehyde. As illustrated in Supplementary Fig. 8, the experimental data indicate linear increases in HCHO concentration within the residence times (<500 ms). Optimal agreement between the experimental and kinetic simulation of HCHO concentration was achieved when the primary yield of HCHO was set to 0.88. The yield of HCHO from ozonolysis of ethene, extensively studied in the literature[4,26,42,45–51], was found to be around 0.9. The primary yield of HCHO employed in our kinetic model aligns well with the previous studies.

To assess the kinetic rates of the reaction between $CH_2OO$ and $SO_2$, the absorption spectra of $CH_2OO$ were measured under different residence times with $SO_2$ introduced at concentrations ranging from $1\times10^{13}$ to $4\times10^{13}$ cm⁻³. As depicted in Supplementary Fig. 9, the concentration of $CH_2OO$ from kinetic simulation was compared to the experimental results spanning 170 to 430 ms. The determined best value of $k(CH_2OO+SO_2)$ was 3.9 ($\pm$0.8) $\times10^{-11}$ cm³ s⁻¹, demonstrating good concordance with the literature values obtained through photolysis synthesis methods[29]. The kinetic data measured in this work suggests that, despite the conjecture that the CIs originated from alkene ozonolysis are produced with higher internal energy[9,26,52] than those arising from photolysis of diiodoalkanes in excess amount of $O_2$[15,53], the reaction rates of the thermally equilibrated CIs are comparable. The constructed reaction network in this study helps validate preceding kinetic investigations. One limitation for the kinetic determinations in this work is that the shortest residence time achieved in the current setup was ~10 ms, making it challenging to capture the full-time profile for rapid $CH_2OO$ reactions. For instance, in the $CH_2OO+SO_2$ reaction, the $CH_2OO$ signal would rise and peak within 5 ms[15,21,54–56]. Achieving a residence time closer to 1 ms would likely require a ~10-fold increase in the pumping speed. Besides, uncertainties in residence time due to flow uniformity or wall losses are expected to be negligible based on the ideal plug flow reactor assessment in Supplementary Table 1, but future studies could further quantify their effects to refine kinetic accuracy.

## Discussion

Formaldehyde oxide intermediate produced in ozonolysis of ethene was directly characterized in situ in the reaction network using CRDS. Measurements of the time profiles of the transient $CH_2OO$ intermediate at various initial reactant concentrations allowed for quantitative assessment of the production and loss processes of $CH_2OO$ in ozonolysis. These $CH_2OO$ concentration profiles benchmarked the modeling of the ethene ozonolysis reaction network and mechanism, determining the branching and various kinetic data of $CH_2OO$ and providing deeper insights into the ozonolysis reaction. The use of flow reactor and laboratory scale conditions provides additional control on mixing and reaction conditions and allows for the study of known secondary reactions in the ozonolysis mechanism, offering an intermediate step between the determination of individual rate constants and the application of reaction mechanisms in environmental chamber studies, which ultimately leads to better atmospheric chemistry modeling.

This direct ozonolysis method could serve as a platform for studying other CIs in larger ozonolysis systems (which may not be readily accessible by photolysis synthesis method). This study also opens door for studying reaction mechanisms of other complex systems. It demonstrates that one needs to be able to measure the difficult but key species, and CIs are the epitome of this. It shows that direct measurements of the time profiles of the key intermediates in the reactions in situ anchor the whole reaction network and can provide greater understanding of the reaction mechanisms.

## Methods

### Flow cell reactor

The flow cell reactor was made of cylindrical quartz tube (57 cm length, 2.2 cm ID, and 2.5 cm OD). The inlets of the flow cell were 0.25–0.38 cm OD quartz tubes, while the outlets were 0.38–2.5 cm OD quartz tubes, used for different flow and pumping speeds. The flows of the reactants were controlled by mass flow controllers (MFCs, Aalborg model GFC17S-EAL6-A0) to maintain continuously stable flow rates. The total pressure inside the flow reactor was precisely monitored by a Cole-Parmer pressure gauge (model EW-68936-00). To achieve different residence times under the same pressure, the total flow rates were varied between 0.4 and 5 sL/min with MFCs of different scales for accurate control. The pumping speed of the vacuum pump (Welch model 1397) was adjusted using an inline valve connected to the flow reactor outlet to keep the pressure constant under different flow rates.

Ethene (99.95%, Matheson) flows were in the 0.1–1.5 sL/min range. Nitrogen (0.2–3 sL/min) worked as a buffer gas to adjust the total pressure and the reactant concentrations. Ozone was generated by an ozone generator (ENALY model 1000BT-12) with an oxygen inlet flow of 0.1–0.4 sL/min. The concentration of ozone in the outlet stream of the ozone generator (before entering the flow cell) was monitored with an ozone monitor (2B Tech model 202) after dilution. By adjusting the voltage of the corona discharge inside the ozone generator, as well as using different flow rates of oxygen inlet, the concentration of ozone before entering the flow reactor was ~2–6%. The flow rate of the ozone and oxygen mixture into the flow reactor was 16–800 smL/min. Ethene and ozone were separately introduced with two different-sized tubes (one inside another) so that they would not mix with each other before entering the flow reactor. At the end of the inlet, the ozone and oxygen mixture in the inner tube (Teflon) encountered the mixture of ethene and nitrogen buffer gas exactly at the entry into the reactor. The ozone concentration inside the flow reactor was further confirmed by measuring its absorption spectra around 330–331 nm using CRDS. To obtain higher ozone concentrations (~10%), a silica gel trap (at −60 °C) was used to trap ozone from a Welsbach ozone generator (model T-408). Compared to the stable ozone concentration using the ozone generator directly, the ozone concentration from the ozone trap gradually decreased upon use and could last only about 2 h. Therefore,

the ozone trap was only used to maximize HCHO features in 370–390 nm at longer residence times (Supplementary Fig. 2). The rest of the HCHO quantification (lower [HCHO]) was performed around 329 nm (Supplementary Fig. 8).

For the studies on the $CH_2OO + SO_2$ reaction (Supplementary Fig. 9), an $SO_2$/nitrogen (~4%) mixture was introduced with a flow rate of 0.2–5 smL/min. The $SO_2$ concentration inside the reactor was measured with its absorption features around 318 nm using CRDS. For the complete scavenging spectra with $SO_2$ (Fig. 2 and Supplementary Fig. 3), a flow rate of 75 smL/min $SO_2/N_2$ mixture was used.

The length of the flow reactor (distance between the inlet and outlet) for most experimental data (residence time > 35 ms) was 57 cm, while shorter flow cell was used to reduce the residence time to ~10 ms. The quantification of the residence times was achieved with the precisely measured flow rates and pressure, and further confirmed by negative injection and pulse injection methods[57].

### Cavity ringdown spectroscopy

A flow cell was used as a plug flow reactor (PFR) to carry out the ozonolysis reaction under different conditions. The reaction cell also had the role of an optical cavity, and the average concentrations of species of interest were measured using CRDS. This CRDS experimental apparatus is shown in Supplementary Fig. 10 and has been used for different ozonolysis reactions[58–60]. Number densities are calculated from the ring-down decay time measurements according to:

$$\alpha = \sum_i \sigma_i(\lambda) N_i + f(\lambda) = \frac{L}{c\ell_s}\left(\frac{1}{\tau} - \frac{1}{\tau_0}\right) \tag{1}$$

where $\sigma_i(\lambda)$ is the absorption cross-section of the $i$-th species at wavelength $\lambda$, $N_i$ is its number density, $f(\lambda)$ is a parametric function to account for broad extinction contributions from the background and unidentified species, $1/\tau_0$ is the decay rate of the empty cell, $1/\tau$ is that of the cell with the sample, $\ell_s$ is the length of the sample in the cell (57 cm in our flow reactor), L is the length of the cell (100 cm), and c is the speed of light.

An Nd: YAG laser (Continuum Surelite II) at 10 Hz was used to pump a tunable dye laser (Lambda Physik ScanMate) to generate red laser radiation from 650 to 680 nm and 726 to 790 nm with different dyes (DCM, pyridine 1, and styryl 8). Then the high-resolution laser radiation (linewidth ~0.13 cm$^{-1}$) was introduced into an Inrad Auto-tracker through a BBO doubling crystal to produce near-UV radiation (linewidth ~0.2 cm$^{-1}$) in the 325–340 nm (for quantifying HCHO[61] and $O_3$) and 363–395 nm (for $CH_2OO$ in Supplementary Fig. 11) wavelength ranges. High reflectivity mirrors (R > 99.95%) centered at 330 and 370 nm (Layertec model 109121 and 109462) were used to obtain ring-down times of 5–13 microseconds. The Nd: YAG pumped dye laser typically scanned the near-UV wavelength at 0.01 nm/step with 20 laser shots for data averaging at each step. The baseline noise of CRDS in this work was about $1–6 \times 10^{-8}$ cm$^{-1}$ ($\Delta\alpha$), depending on the wavelength used and instrumental conditions. This corresponds to an estimated limit of detection of $CH_2OO$ of ~$1 \times 10^{10}$ cm$^{-3}$ (at 363–395 nm where effective path length of empty cell is ~4000 m and typical ringdown time is ~13 microseconds), based on the reference absorption cross sections from Foreman et al.[34] in the MPI-Mainz UV database[62]. The detection limit of HCHO in Supplementary Fig. 8 was estimated to be ~$1 \times 10^{12}$ cm$^{-3}$ from the baseline noise (~$6 \times 10^{-8}$ cm$^{-1}$) and the 328–330 nm absorption cross sections (~$6 \times 10^{-20}$ cm$^2$) from the HCHO reference rovibronic spectra[61].

Reactant concentrations were varied from $9 \times 10^{14}–1.8 \times 10^{15}$ cm$^{-3}$ for ozone and $5 \times 10^{16}–1 \times 10^{17}$ cm$^{-3}$ for ethene. Given the slow production of CI (k(ethene + $O_3$) = $1.6 \times 10^{-18}$ cm$^3$ s$^{-1}$) and the existence of several depletion processes (decomposition, reactions with ethene, ozone, formaldehyde, etc.), short residence times were used to inhibit secondary reactions as much as possible. Our experimental conditions

allowed for residence times ranging from 10 to 500 ms, leading to estimated $CH_2OO$ concentrations of $1–3.5\times10^{11}\,cm^{-3}$, lower than or close to what would be generated by photolysis of diiodomethane and reaction with oxygen $(10^{11}–10^{13}\,cm^{-3})$[21,34,35,41,43,44]. The change in ring-down decay rate $\Delta(1/\tau) = (1/\tau) - (1/\tau_0)$ was measured from 363–395 nm and compared to the reference $CH_2OO$ absorption cross sections. The average number density can be determined using Equation 1, provided that there exist distinct absorption features of $CH_2OO$.

### Kinetic simulation

A kinetic model was constructed, and the software package KINTECUS[63] was used to estimate concentrations of each CSTR by integration of the system of ordinary differential equations with a Bulirsh-Stoer method. Briefly, it comprises the ozonolysis reaction[4,11], decomposition of the "hot" CI[42], reactions of the "stabilized" CI[21–25,39,64–68], $HO_x$ chemistry[42,69,70], radical reactions with ozone and ethene, and reactions of the $HOCH_2CH_2$ radical produced by the reaction of OH with ethene.

The mechanism of ozonolysis of ethene was constructed based on our previous work[28] and expanded to include more secondary chemistry with existing and recent kinetic data from the literature[4,21,22,24,25,39,64,69–71]. Supplementary Table 2 shows the mechanism used to model the ozonolysis reaction. From the ozonolysis reaction (R1), the branching ratios of the "stabilized" and "hot" Criegee intermediates are explicitly set according to Hatakeyama et al.[11] and Yang et al.[28]. The "hot" $CH_2OO$ will readily decompose and isomerize according to Copeland et al.[42] and references therein, producing H atom, OH radicals, and other fragments (R2–6). The OH radical and H atom will be involved in chemistry with $O_3$ and the reactions are included in the model with rate constants from evaluated kinetic data and from Copeland et al.[42,69,70] (R7–12). The chemistry of H atom and OH radicals with ethylene has been included with rate constants from evaluated kinetic data as well (R13–19). The stabilized fraction of $CH_2OO$ undergoes several bimolecular reactions, as well as unimolecular decomposition. Kinetic studies have been performed with stabilized $CH_2OO$ produced by photolysis of $CH_2I_2$ and subsequent reaction with $O_2$ by several groups[21–25,64]. In the model, those reactions involving products from ethene ozonolysis are added (R20–33). The three major consumption reactions of $CH_2OO$ (with ethene, $O_3$, and HCHO) are studied by comparing with experimental data while varying corresponding rate constants, and the best-fitting results are presented. The reaction rates of $CH_2OO$ with ethene, $O_3$, and HCHO have been directly determined with the photolysis method and these kinetic data were used as starting points for fitting the model. The concentrations of all the species in the model were free running after initialization, while the reaction rate constants were varied between different runs. Stabilized $CH_2OO$ can also undergo unimolecular processes and are described in the model based on data from previous studies[21,54,72–74] and the evaluated kinetic data[70]. The ozonolysis reaction produces water and, while the reaction of $CH_2OO$ with water is slow ($k < 5\times10^{-16}\,cm^3\,s^{-1}$), the rate of $CH_2OO$ with water dimer is much faster ($k = 6.2\times10^{-12}\,cm^3\,s^{-1}$). However, from the equilibrium data between water and water dimer[65,66], it was concluded that water dimer formation in the system was not significant. Products of the reaction of $CH_2OO$ with carbonyls and organic acids were obtained from experimental and theoretical information in the literature[21,22,64,67,68]. The reaction of $CH_2OO$ with $SO_2$ was added for modeling $SO_2$ titration experiments to get the reaction rate constant (Supplementary Fig. 9). Ketohydroperoxide (KHP) was recently found to be a minor pathway existing at the starting stage of ozonolysis[51] and its chemistry is also included (R1, R80–82). $HO_x$ plays a role in the ozonolysis mechanism either by having an effect on the $HO_x$ budget or by reacting with secondary products of ozonolysis such as formaldehyde or acetaldehyde. This chemistry is included in the model (R34–45, R54–65) with rate constants from evaluated kinetic data[69,70]. Using an ozone generator to produce ozone via corona discharge in a stream of pure oxygen results in high concentrations of oxygen going into the system. Therefore, the secondary chemistry of oxygen has been included in the kinetic model (R46–49, R66–79). The reactions of $HO_x$ with products of ozonolysis indicate some vinoxy radical production that is expected to be minimal. Nevertheless, vinoxy radical and oxygen reactions were included in the model for completeness (R50–53). Oxygen mainly acts as a scavenger of ozonolysis products, particularly the $C_2H_5$ radical and $HOCH_2CH_2$ radical (R67–79). Supplementary Table 2 shows the mechanism used to model the ozonolysis reaction. These reactions are translated into a system of ordinary differential equations (ODEs) and integrated using a Bulirsh-Stoer method by the software package KINTECUS[63]. As the dimensionless parameters shown in Supplementary Table 1 comply with the criteria by Cutler et al.[75] and Lee et al.[76] for non-Poiseuille flow, the reactor was assumed to behave reasonably close to a plug-flow reactor and modeled as a series of continuously-stirred tank reactors (CSTRs) in tandem, where the output of the previous CSTR becomes the input of a new CSTR along the flow cell. A total of 10 CSTRs were used to simulate a concentration profile along the flow cell. The use of additional segments showed convergence in the concentration profiles and, thus, only ten segments were used to facilitate simulations. A concentration profile of $CH_2OO$ along the flow cell at different residence times is shown in Supplementary Fig. 1.

### Reporting summary

Further information on research design is available in the Nature Portfolio Reporting Summary linked to this article.

## Data availability

Source data are provided with this paper.

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

## Acknowledgements

This work was supported by the US National Science Foundation grant CHE-2155232. M.C. acknowledges support from UCMEXUS-CONACYT Doctoral Fellowship. L.Y. acknowledges support from UC Riverside Dissertation Research Grant. J.Z. acknowledges support from UC Riverside OASIS Internal Funding Award.

## Author contributions

All authors conceived the project. M.C. and L.Y. developed the methodology. L.Y. and M.C. conducted experiments and kinetic simulation and performed data analysis. J.Z. supervised the project. All authors wrote the manuscript.

## Competing interests

The authors declare no competing interests.
