## [Transparent Peer Review file · Nature Communications]

Direct Measurement of the Criegee Intermediate CH₂OO in Ozonolysis of Ethene

Corresponding Author: Professor Jingsong Zhang

Version 0:

Reviewer comments:

Reviewer #1

(Remarks to the Author)

The authors present direct observations of the Criegee intermediate formaldehyde oxide (CH₂OO) formed during the ozonolysis of ethene made using cavity ringdown spectroscopy in a flow cell reactor. Time-resolved concentrations of CH₂OO in the flow reactor could be determined from knowledge of the absorption cross-sections for CH₂OO, enabling investigation of the stabilized Criegee intermediate (SCI) yield and reaction kinetics. Measurement of CH₂OO + SO₂ kinetics, which are now relatively well-established in the literature, was used to confirm the spectral assignment of CH₂OO.

Direct measurements of Criegee intermediates in ozonolysis reactions have been a goal of atmospheric scientists for a number of years, and the need for direct measurements has become more apparent in recent years as our knowledge of Criegee intermediate chemistry has been challenged by studies making use of photolytic Criegee intermediate precursors. Direct measurements in ozonolysis reactions represent a number of challenges, and the authors have achieved a significant goal in achieving the measurements described in the manuscript. However, there are some aspects of the analysis and discussion of previous work that detract from the significant achievements in experimental observations made by the authors. I have a number of comments on the manuscript, detailed below, that should be addressed prior to publication.

Line 26: 'Products of ozonolysis ultimately...' should be changed to 'Products of ozonolysis can ultimately...', it is not always the case that the products described will be formed, and there are significant uncertainties in some of the mechanisms forming the products listed, particularly so for HOMs.

Lines 43-45: References should be given.

Lines 46-50 (and elsewhere): There should be some comment on behaviour at atmospheric pressure, and current IUPAC/NASA-JPL recommendations for SCI yields. The current IUPAC recommendation for the OH yield is (17 ± 5) %.

Lines 53-58: What is the anticipated timescale for stabilization of Criegee intermediates produced by ozonolysis? Do theory studies support the argument that there might be significant bimolecular reactions of excited Criegee intermediates? It would be helpful to elaborate on the comments made here and to provide details of reactions/processes that show discrepancies between photolytic measurements and studies of ozonolysis reactions.

Line 71: Please quantify 'very long time signal integration' and 'poorly-defined reaction time' for comparison with the current work.

Line 75 and line 95: There are a number of literature references for CH₂OO absorption cross-sections that show similar spectra, but with better signal-to-noise ratios than the data reported by Foreman et al. used for comparison with the current work. These should be included in the manuscript for a fair assessment of the signal-to-noise obtained in the spectra obtained in the current work. It is not clear that the higher resolution used in this work has led to the improved signal-to-noise compared to Foreman et al. as others with lower resolution also have better signal-to-noise ratios.

Line 98: What are the detection limits for CH₂OO and HCHO?

Line 101: The absorption cross-sections for HCHO, and other products, are orders of magnitude lower than those for

CH₂OO. Lack of observations of reaction products does not preclude these species being present at concentrations that might impact CH₂OO.

Line 114: It seems more appropriate to describe the estimates of rate constants as measurements.

Line 117: Please provide a table in the extended data/supplementary information giving initial conditions for all experiments performed. The pressures at which experiments were performed should also be clearly stated in the manuscript.

Lines 120-125 and 137-142: There is some repetition here. It seems unnecessary to explain how the balance between production and consumption leads to the observed profiles.

Line 122 (and elsewhere): The unimolecular decomposition of stabilized CH₂OO has been shown to be $\ll 1$ s⁻¹ under the conditions employed in the current work. It is unlikely the unimolecular decomposition of the SCI is playing any significant role.

Line 130-135: The more common way to model the observations would be to simply start with the initial concentrations for ethene and ozone and run the model forwards in time. If the flow reactor is under plug flow conditions with negligible axial diffusion (as outlined in the extended data), the simple model should be expected to give similar results? What are the benefits of using the model approach described and how do the results differ from the more simple model?

Line 142: It would be helpful to plot the percentage contributions to CH₂OO loss as a function of time (perhaps in the extended data or supplementary information).

Line 147: What is the main source of HCOOH? It is included in the table in the supplementary information as one of several products of CH₂OO + HCHO, in Figure 4 with a different yield, and not in the extended data Table 2 (see also comments below on the rate constants given in the extended data Table 2).

Line 172: The paper by Vereecken et al. discussed in the manuscript does not report their latest findings on the reaction between CH₂OO + O₃. While the paper mentioned did predict a rate constant of 1×10^{-12} cm³ s⁻¹, later work by the same group (2015) using a higher level of theory revised their prediction to 4×10^{-13} cm³ s⁻¹. The work by Chang et al. has also been shown to be impacted by secondary chemistry involving CH₂OO + IO. Here, and in several other places in the manuscript (references to spectra by Foreman et al., product yields for CH₂OO + HCHO, results of Copeland et al. for example) the authors are a little too selective in the previous data they choose to discuss.

Lines 174, 178, 184 (and elsewhere): Where results of previous work are given, uncertainties should be included throughout.

Line 184: Luo et al. also reported product yields for CH₂OO + HCHO at similar pressures to those used in the current work. Are these values used in the model? HCOOH is an important potential product of the reaction, and reacts rapidly with CH₂OO.

Line 198: Can the results of the current work be used to say anything about expected SCI yields at atmospheric pressure?

Line 205: The yield of HCHO is usually given as 1, owing to its co-production in the initial ozonolysis reaction. The Copeland et al. study features heavily in the manuscript, but there is an extensive body of work in this area that has been evaluated by IUPAC and others. How do these compare?

Line 213-214: It would be helpful to provide some references to the conjecture mentioned.

Line 282: A diagram of the experimental setup would be helpful.

Line 298: Please give the effective path length and typical values for the ringdown time of the empty cell.

Lines 305 and 306: Concentrations on the order of 10¹¹ cm⁻³ have been reported for CH₂OO in photolytic experiments. The statement that the concentrations in the current work are orders of magnitude lower than those in previous photolytic experiments is not true.

Line 319: How significant are the reactions in the model? The mechanism is comprehensive, but do all the reactions listed need to be included? What are the key reactions in the model? There are differences between the reactions, rate coefficients, and product yields between the table in the supplementary information and the extended data (Table 2) which should be clarified and explained.

Line 333: Are concentrations of O₃, C₂H₄, and HCHO in the model constrained to observations or free-running after initialisation? If free-running, how do they compare to observations?

Line 336: Kinetics of unimolecular processes are only estimated by Chhantyal-Pun et al. More detailed studies have been performed which show the unimolecular decomposition of the SCI to be insignificant under the conditions of the current work.

Line 533 (and elsewhere): The primary reference for the data should be given in place of (or in addition to) the reference to the MPI-Mainz UV/Vis Spectral atlas.

Figure 3: What is the difference between the open symbols and the dashed lines? The dashed lines appear to be from the simulations but do not go through all the open symbols. Can uncertainties be estimated for all the experimental data?

Extended data Figure 1 (and elsewhere): In some places, units include “molecules” and in other places they do not. Either should be acceptable, but there should be consistency.

Extended data Figure 2: The reference spectrum appears to be at a much lower resolution than the observations. A reference spectrum with similar resolution should be used. Effective absorption cross-sections for HCHO are likely impacted by the spectral resolution, and as such the concentrations of HCHO reported throughout may be impacted, with potential consequences for the conclusions regarding the importance of CH₂OO + HCHO. What other species might be expected to be present and to contribute to the broad absorption?

Extended data Figure 5: It's not clear that there are any differences between the fits. It would be helpful to tabulate the values for the SCI yield and rate constants derived from the fits shown in the different panels. Do the results from the different fits agree?

Extended data Figure 6: Uncertainties should ideally be estimated for all experimental data points. What are the implications of the results for the SCI yield at atmospheric pressure?

Extended data Figure 8: The caption to the figure gives the same initial ozone and ethene concentrations for all data, but the observed and modelled data for CH₂OO at [SO₂] = 4 × 10¹³ cm⁻³ are higher than those at [SO₂] = 2 × 10¹³ cm⁻³. Please clarify. How do the traces compare at earlier times? It would be better to show the full time series.

Extended data Table 2: Are the values reported the results of the fits? The differences with the mechanism given in the supplementary information are not clear. The rate constant for CH₂OO + HCOOH is two orders of magnitude slower than has been reported in the literature – why? Why are no products specified?

Reviewer #2

(Remarks to the Author)

This manuscript describes direct in situ measurement of formaldehyde oxide Criegee intermediate (CH₂OO) in the ozonolysis of ethene, with sufficient effective time resolution and fidelity to validate explicit models of the CH₂OO chemistry. That is a substantial achievement that will have a significant impact in understanding the details of this critical prototype ozonolysis experiment. However, there are some aspects of the work that require clarification before publication.

Uncertainties. The demonstration reaction for the new capability is the determination of the CH₂OO stabilization yield in ethene ozonolysis. The reported yields are in reasonable agreement with previous results, but the uncertainty estimates in the present work are really surprisingly small (relative uncertainties of 4%). I think these estimates must not be the absolute uncertainties and need to be explained more clearly. For example, how does uncertainty in the absorption cross section for CH₂OO contribute to the uncertainty in yield? Naïvely I would think it would directly contribute, as the concentration is determined from the absorption. Yet the stated uncertainty in the yield is at least 5 times smaller than that of the reference cross section (the Foreman et al. (ref 31) measurements referenced in the manuscript have a 1σ uncertainty of ~28%, according to the table linked from the Mainz database (ref 44)). Moreover, even the uncertainty inherent in the kinetic model seems like it has to be significantly more than 4%. I imagine the actual absolute uncertainty is similar to this group's previous measurements (ref 26).

Kinetics. One of the key strengths of this work is the in situ probing of carbonyl oxide in a manner that allows quantitative comparisons to kinetic models. In that regard, a clearer description of the experiment and the constraints on the time response would be valuable. As I gather from schematic figures in previous work (references 42 & 43) and the description provided here, the CRDS probe is along the flow axis.

--- I think that means the measurement (even under the simplified flow assumptions employed) integrates from injection up to the nominal total residence time, so for a given flow rate the measurements is essentially probing an average of a horizontal slice in Extended Data Figure 1. It would be valuable if this were more clearly described. This will guide future researchers in reactor design.

--- The description of the concentration profiles could be clearer, based on what I think the measurement probes. Some of the wording (“rapidly measure kinetics” (line 73-74) or “complete transient time profiles” (in the abstract line 19, and in line 78)) sounds like a time-resolved measurement rather than a quasi-steady state measurement that probes a tunable range of reaction times. This should be stated more accurately throughout.

--- I think that the demonstration that CH₂OO can be quantitatively probed in an ozonolysis system, in a manner that detailed kinetics can be followed, is the main contribution of the paper, much more important than the specific kinetics measurements. The authors show that suggest that the scavenging of CH₂OO by SO₂ can be quantified (Extended Data Fig. 8). The accuracy of the kinetics is always affected by some of the flow reactor design choices – subsequent work will build on this and improve the kinetics fidelity, if the limitations are clearly described. If the authors could share an analysis of the major limitations on the kinetics determinations, that would improve the impact greatly.

Reviewer #3

(Remarks to the Author)

Dear Editor:

This work utilized the high sensitivity of cavity ring down spectroscopy to directly probe the simplest Criegee intermediate CH₂OO in the ozonolysis reaction of C₂H₄, the most fundamental ozonolysis reaction. To my knowledge, this is the first work of in-situ spectroscopic probe of CH₂OO in the ozonolysis reaction.

Due to the high exothermicity of primary ozonide decomposition, the Criegee intermediates produced in ozonolysis reaction may have high internal energy, very different from the Criegee intermediates produced from photolysis of diiodoalkanes. Thus, it is necessary to study the ozonolysis reaction itself. Direct observation and kinetic measurements of Criegee intermediates in ozonolysis in situ in real time will anchor the reaction mechanisms and greatly improve our fundamental understanding of the whole reaction network.

In this work, the signal-to-noise ratio of the reported cavity ring down spectroscopy of CH₂OO is very nice, not only giving confidence that CH₂OO has been truly probed, but also offering the best version of the UV spectrum of CH₂OO.

The absolute concentration of CH₂OO was measured in the reaction, offering a valuable check of the kinetics of the ozonolysis reaction.

I have a few minor comments as listed below. Further review is not needed.

1) It will be nice if a schematic plot of the flow reactor is given in the SI.

2) The authors use the term of "reaction time" throughout this manuscript. But after reading it carefully, I think the "reaction time" should be actually "residence time". For a given residence time, say 140 ms, the reaction time actually spanned from 0 to 140 ms. Thus, the use of "reaction time" is confusing and potentially MISLEADING. Please clarify this issue when it first appears in the manuscript and use a proper terminology. Related terms like "the transient time profiles of CH₂OO" should also be defined carefully. There is a similar issue for Figure 3.

3) Page 3: "The higher spectral resolution in this work (0.01 nm) compared to the reference (~0.12 nm) assures an improved signal-to-noise ratio, allowing more accurate determination of CH₂OO concentration using the vibronic band features spaced by ~600 cm⁻¹ with half-peak widths of ~200 cm⁻¹ in the following kinetic experiments." I do not agree. The higher spectral resolution is different from the improved signal-to-noise ratio. Although both the spectral resolution and the signal-to-noise ratio are greatly improved in this work. They are different aspects. For such broad vibronic bands, a high spectral resolution will not help. In addition, it is a bit hard to compare "nm" to "cm⁻¹". Please use the same unit for this paragraph (at least, added in parenthesis).

4) Figure 1 and page 8: "... based on the reference absorption cross sections from Foreman et al." There are other works that also reported the absolute absorption cross sections of CH₂OO. They must not be ignored, especially those with a higher accuracy. In particular, Foreman et al. stated that "The uncertainty in the number density (~28%) estimate is easily the largest uncertainty in the absorption cross section determination." This uncertainty is not small.

5) Extended Data Fig. 1: Should "CRDS Segment" in the x-axis be "CSTR Segment"
It seems that this figure is an interpolated one (for the CH₂OO concentration between each segments). In my opinion, it is better to show the original results of those 10 CSTR segments without doing any interpolation, although it may look less pretty. If interpolation is really necessary, it should be explained how the interpolation was done.

6) Page 7: The authors use "L/min" and "mL/min", which I believe should be "sL/min" and "smL/min" (the standard mass flow at 273 K and 760 Torr). Please clarify this.

7) Please give company names and model numbers (a lot are missing in the current version) of all the important instruments used.

8) Others.

page 2, please modify "For many decades, ..." to "For decades, ..."

page 3, please modify "... to prevent a high production of reaction products along the reaction cell." to "... to prevent a high production of reaction byproducts along the reaction cell."

page 5, please modify "as more formaldehyde oxide remains" to "as more CH₂OO remains".

Version 1:

Reviewer comments:

Reviewer #1

(Remarks to the Author)

The authors have clarified most of the points raised in the previous reviews which has improved the manuscript. However, there remain a small number of areas which should be addressed more completely prior to publication.

The previous comment on the timescale for stabilization of Criegee intermediates, likelihood of bimolecular reactions of 'hot' Criegee intermediates, and discrepancies between photolytic measurements and studies of ozonolysis reactions (lines 53-58) has not been satisfactorily addressed. The timescales referred to in the response do not relate to stabilization of the nascent Cl, and the studies of bimolecular reactions of 'hot' species refer to very different systems. This should be more explicit in the manuscript, which gives the impression that there is evidence for bimolecular reactions of 'hot' Criegee intermediates in ozonolysis reactions. If such reactions were to occur, faster reactions of Criegee intermediates would be reported in ozonolysis studies compared to photolytic studies – if anything, it is the other way around.

The responses also still suggest that higher spectral resolution leads to improved signal to noise. This is not the case. The authors are correct that more data points improve the signal to noise, but this is entirely separate from the spectral resolution. An experiment with a greater number of data points, each with lower spectral resolution than the current work, would also improve the signal to noise. The resolution used in the current study does not better capture the peak and valley features in the CH₂OO spectrum as the spacing between them is such that they can be captured with lower resolution measurements.

It is incorrect to state "both the higher spectral resolution in this work (0.01 nm) compared to the references (~0.12-2 nm) and the improved signal-to-noise ratio allowed more accurate determination of CH₂OO concentration...". Fundamentally, the determination of CH₂OO concentrations in this work is reliant on absorption cross-sections reported in the previous, lower resolution, studies, and it is not possible that the current work can both use those results and report more accurate concentrations.

The authors also refer to "uncertainties in of the cross-sections determined in this work", but no cross-sections have been determined. This requires correction/clarification.

In the responses, the authors have clarified the question regarding the yield of HCOOH from CH₂OO + HCHO in the model, but Extended Data Table 2 shows the total rate coefficient for CH₂OO + HCHO and it could be clearer in the manuscript what branching ratios have been used in the model for this reaction.

In several places, the authors refer to "complete time profiles", but this is not an entirely accurate description as there not a return to zero concentrations and elsewhere the authors refer to the challenges capturing "the full time profile".

Uncertainties in values reported in previous work have been added in places, but there are still values referred to in the manuscript without uncertainties. This should be corrected throughout.

Reviewer #2

(Remarks to the Author)

I have read the response to my comments and those of the other reviewers and I appreciate the authors' careful consideration of all the suggestions. The revised manuscript is considerably stronger and better highlights the significant results of this work. One minor remaining comment, which I think the authors need not address because it does not affect the end result, is that I do not believe that the uncertainty in the kinetic modeling can be as low as 4% as the authors state. However, I do quite easily accept that other sources of error dominate and that the final error estimates in the manuscript are reasonable.

Reviewer #3

(Remarks to the Author)

Dear Editor:

I feel that the authors have addressed the concerns raised by the reviewers and this work is publishable now. Just a couple of minor suggestions.

1) For the absolute values of CH₂OO cross sections, both JPL and IUPAC are using the results of Ting et al. See [https://uv-vis-spectral-atlas-mainz.org/uvvis/cross_sections/Organics\(carbonyls\)/Carbonyl oxides/CH₂OO.spc](https://uv-vis-spectral-atlas-mainz.org/uvvis/cross_sections/Organics(carbonyls)/Carbonyl%20oxides/CH2OO.spc)

In addition, the error of Ting et al., $(1.23 \pm 0.18) \times 10^{-17} \text{ cm}^2$ at 340 nm, 15%, is smaller than that of Foreman et al.

Thus, it is better to use Ting's values for the concentration determination.

In addition, lines 105–109, "The relative uncertainties in CH₂OO absorption cross sections in the references listed (20–30%^{17,38,39}) ..." should be revised accordingly.

2) The authors mentioned in their Reply Letter: "Our high-resolution spectra are from one scan without any spectral

averaging."

Does this mean that one data point shown in the figure was from the result of a single laser shot?

This should be clarified in the text such that if another group wants to repeat the experiment, they can know how many laser shots are needed.

Further review is not needed.

Version 2:

Reviewer comments:

Reviewer #1

(Remarks to the Author)

The authors have addressed the comments raised during the reviews and their responses are appreciated. The manuscript is much improved and the authors should be commended for their efforts in the measurements reported.

Dear Editor:

We thank all the reviewers for their comments on the manuscript for its improvements. We have incorporated the reviewers' comments to enhance the manuscript's correctness, completeness, readability, and conciseness. The Excel file entitled "Source Data", which contains the raw data for all display items, with each figure and table provided in separate sheets, is also uploaded.

Point-by-point responses to the reviewers' comments are as follows.

Reviewer #1 (Remarks to the Author):

The authors present direct observations of the Criegee intermediate formaldehyde oxide (CH₂OO) formed during the ozonolysis of ethene made using cavity ringdown spectroscopy in a flow cell reactor. Time-resolved concentrations of CH₂OO in the flow reactor could be determined from knowledge of the absorption cross-sections for CH₂OO, enabling investigation of the stabilized Criegee intermediate (SCI) yield and reaction kinetics. Measurement of CH₂OO + SO₂ kinetics, which are now relatively well-established in the literature, was used to confirm the spectral assignment of CH₂OO.

Direct measurements of Criegee intermediates in ozonolysis reactions have been a goal of atmospheric scientists for a number of years, and the need for direct measurements has become more apparent in recent years as our knowledge of Criegee intermediate chemistry has been challenged by studies making use of photolytic Criegee intermediate precursors. Direct measurements in ozonolysis reactions represent a number of challenges, and the authors have achieved a significant goal in achieving the measurements described in the manuscript. However, there are some aspects of the analysis and discussion of previous work that detract from the significant achievements in experimental observations made by the authors. I have a number of comments on the manuscript, detailed below, that should be addressed prior to publication.

Line 26: 'Products of ozonolysis ultimately...' should be changed to 'Products of ozonolysis can ultimately...', it is not always the case that the products described will be formed, and there are significant uncertainties in some of the mechanisms forming the products listed, particularly so for HOMs.

Response: Revision was added to line 26 (*note that the line number refers to that in the revised manuscript text with track changes, and the same thereafter*), "Products of ozonolysis can ultimately lead to the production of highly oxidized molecules (HOMs)..."

Lines 43-45: References should be given.

Response: Citations were added in lines 43–46.

Lines 46-50 (and elsewhere): There should be some comment on behaviour at atmospheric pressure, and current IUPAC/NASA-JPL recommendations for SCI yields. The current IUPAC recommendation for the OH yield is $(17 \pm 5) \%$.

Response: The IUPAC-recommended sCI yields at atmospheric pressure, OH yields, and corresponding citations were added to lines 48–49, "...the nascent yield of "stabilized" CH₂OO is $\sim 0.20^{11,12,26}$, while it increases to ~ 0.42 at atmospheric pressure due to collisional stabilization.²⁷ In addition, its low OH yield (~ 0.17)^{27,28} is strong evidence..."

Lines 53-58: What is the anticipated timescale for stabilization of Criegee intermediates produced by ozonolysis? Do theory studies support the argument that there might be significant bimolecular reactions of excited Criegee intermediates? It would be helpful to elaborate on the comments made here and to provide details of reactions/processes that show discrepancies between photolytic measurements and studies of ozonolysis reactions.

Response: The corresponding comments and citations have been revised in lines 56-57.

The anticipated timescale of sCI production: Stabilized CIs can be produced both from collisional stabilization (at atmospheric pressure) and directly from ozonolysis (nascent sCIs born with less energy (see Nguyen et al.¹ and Yang et al.²). For the latter, the timescale of nascent sCI or hot CI production from POZ cleavage in ethene ozonolysis is on the order of several ps (Pfeifle et al.³), yet the formation of POZ from the ethene + O₃ 1,3-cycloaddition reaction is the rate determining step ($1.60 \times 10^{-18} \text{ cm}^3 \text{ s}^{-1}$).

Only a few bimolecular reactions of nonthermal species have been investigated very recently (Klippenstein et al.⁴). Jasper et al.⁵, Burke et al.⁶, Plane and Robertson⁷ investigated nonthermal bimolecular reactions of CH₄ + OH/H/O₂, CH₂O + O₂, and formation of metal silicates, respectively, and showed that such hot reactions can lead to an enhanced reaction rate by orders of magnitude. As Klippenstein⁴ suggested, “The preparation of molecules with an excess of energy in their internal states can also lead to an enhancement of their bimolecular reaction rates. ... Thus, any consideration of the effect of prompt dissociation should also consider the increase in the species bimolecular reaction rates⁴.” Although such trajectory calculations have not been reported to study the bimolecular reactions of hot CIs to the authors’ knowledge, it is reasonable to anticipate similar enhancement in their hot bimolecular reaction rates than sCIs. The open question is whether such enhanced bimolecular reaction rates of CIs produced in ozonolysis are high enough to be competitive with the fast unimolecular decomposition of hot CIs.

Line 71: Please quantify ‘very long time signal integration’ and ‘poorly-defined reaction time’ for comparison with the current work.

Response: The details of the cited work by Womack et al.⁸, signal integration “4.3 hours or 93000 sample injections”, and their residence time “estimated to be <0.5 s with a 6-Hz sampling rate”, were added to lines 72–73.

Line 75 and line 95: There are a number of literature references for CH₂OO absorption cross-sections that show similar spectra, but with better signal-to-noise ratios than the data reported by Foreman et al. used for comparison with the current work. These should be included in the manuscript for a fair assessment of the signal-to-noise obtained in the spectra obtained in the current work. It is not clear that the higher resolution used in this work has led to the improved signal-to-noise compared to Foreman et al. as others with lower resolution also have better signal-to-noise ratios.

Response: The available literature references are compared in the revised Figure 1, lines 90-91 (“...in comparison with CH₂OO reference spectra by Foreman et al.”), and supplementary Figure S2. In addition to Foreman et al.⁹, the spectra of Ting et al.¹¹ and Mir et al.¹² were added in Figure 1 for comparison. Compared with Foreman et al.⁹, our CRDS spectra have better signal-to-noise ratio (thus higher precision and accuracy) and better resolution (0.01 nm vs. 0.12 nm, and although both can well represent the diffuse vibronic spectral features that have ~8 nm spacings and ~3.5 nm half-peak widths, our spectra have more data points to fit the spectra for better precision and accuracy). Compared with Ting et al.¹¹ and Mir et al.¹², our spectra probably have similarly good signal-to-noise ratio (more discussion on their signal-to-noise ratios in the following), but have much higher spectral resolution that can better represent

the diffuse vibronic spectral features (hence, with the similar signal-to-noise ratio, our spectra capture more accurately the peak and valley spectral features – similar to “oversampling” – and have many more data points to fit the spectra for better precision and accuracy). Therefore, both the high spectral resolution and the very good signal-to-noise ratio in our spectra contribute to the accuracy of concentration measurements (even for the relatively broad, diffuse vibronic spectra). We have revised the discussion in lines 101-105, “Both the higher spectral resolution in this work (0.01 nm) compared to the references ($\sim 0.12 - 2$ nm) and the improved signal-to-noise ratio allowed more accurate determination of CH₂OO concentration using the vibronic band features spaced by ~ 8 nm (or 600 cm^{-1}) with half-peak widths of ~ 3.5 nm (or 200 cm^{-1}) in the following kinetic experiments.”

Further considerations are as follows. Higher spectral resolution tends to lead to higher noise levels due to more data points and spectral details, instead of improving signal-to-noise ratio. In contrast, fewer data points or lower-resolution spectra may “help” smooth and improve the appearance of “signal-to-noise ratios”. As shown in supplementary Figure S2, Foreman et al.⁹ spectra are interpolated to 1-nm resolution (no smoothing) and compared with their original spectra (0.12 nm) and other references. The interpolated spectra of Foreman et al.⁹ appear to have similar “signal-to-noise” ratio with the lower resolution spectra reported by Sheps et al. (2 nm)¹⁰, Ting et al. (2 nm)¹¹, and Mir et al. (1.5 nm)¹². In another example, “Figure 2(b)” in Ting et al.¹¹, where they showed their raw data in the grey shades, had similar “signal-to-noise ratios” with the original data of Foreman et al.⁹ The reason that the references of Ting et al.¹¹ (or other spectra) look smoother than Foreman et al.⁹ is because they were done with signal averaging (Ting et al.¹¹ did average of 99 spectra) or obtained in low resolution (10-times larger x-axis intervals) with fewer data points, instead of apparently having “better signal-to-noise ratios”.

It should be noted that in Figure 1, both our spectra and Foreman’s spectra⁹ present raw spectra without any smoothing treatment, to avoid the loss of any spectral details. Our high-resolution spectra are from one scan without any spectral averaging. Further, the signal-to-noise ratio achieved in our spectra is very good. Because the diffuse vibronic bands of CH₂OO are spaced by approximately 8 nm and have half-peak widths of ~ 3.5 nm, *spectral resolution better than 2-nm, in combination with good signal-to-noise ratio, should be more reliable for quantifying these features.*

In comparison with the literature references, our CRDS spectral resolution of ~ 0.01 nm is the highest. Furthermore, our CRDS spectra also have very good signal-to-noise ratio. The very good signal-to-noise ratio of our CRDS spectra is probably because (1) the method we generate CH₂OO is under much milder conditions: ozonolysis reactions are stable as long as flows and pressure are stable (controlled by mass flow controller and pump), unlike laser-triggered photolysis reactions, and (2) there are little or no interference from other species in this spectra range: the largest interference in this wavelength range comes from HCHO, but is negligible for our method when residence time is shorter than 0.5 s, while photolysis method can generate other species like IO radicals that have stronger rovibronic peaks around 400 nm.

In summary, Sheps et al.¹⁰, Ting et al.¹¹, Mir et al.¹² and Foreman et al.⁹ are the available spectra with enough resolution to show the vibronic bands of CH₂OO, with their spectral resolution being 2 nm, 2 nm, 1.5 nm and 0.12 nm, respectively. The spectra by Sheps et al.¹⁰ and Beames et al.¹³ may have overestimated the cross sections as discussed by Ting et al.¹¹, Foreman et al.⁹ and Mir et al.¹², and therefore scaled by 0.3 in supplementary Figure S2, as suggested by Ting et al.¹¹ and Mir et al.¹². In comparison, our CRDS spectra have both the highest spectral resolution and the very good signal-to-noise ratio, with both contributing to better accuracy of concentration measurements (even for the relatively broad, diffuse vibronic spectra).

Line 98: What are the detection limits for CH₂OO and HCHO?

Response: The detection limit of CH₂OO was detailed in the Methods section in lines 331-333, “The baseline noise of CRDS in this work was about $1 - 6 \times 10^{-8} \text{ cm}^{-1}$ ($\Delta\alpha$), depending on the wavelength used and instrumental conditions. This corresponds to an estimated CH₂OO detection limit of $\sim 1 \times 10^{10} \text{ cm}^{-3}$.”

The detection limit of HCHO has been added to lines 335-338, “The detection limit of HCHO in Extended Data Fig. 7 was estimated to be $\sim 1 \times 10^{12} \text{ cm}^{-3}$ from the baseline noise ($\sim 6 \times 10^{-8} \text{ cm}^{-1}$) and the 328.8 – 330 nm absorption cross sections ($\sim 6 \times 10^{-20} \text{ cm}^2$) from the HCHO reference rovibronic spectra (Smith et al.¹⁴).” The measurements of HCHO shown in Extended Data Fig. 7 used the 328.8 – 330 nm wavelength range. It should be noted that, in the different range of 363 – 395 nm where CH₂OO was measured, the HCHO peak cross section is only $\sim 3.4 \times 10^{-22} \text{ cm}^2$, and thus, HCHO concentrations lower than $2 \times 10^{14} \text{ cm}^{-3}$ are not detected in this range.

Line 101: The absorption cross-sections for HCHO, and other products, are orders of magnitude lower than those for CH₂OO. Lack of observations of reaction products does not preclude these species being present at concentrations that might impact CH₂OO.

Response: Clarification and revision have been made to lines 109-116: “HCHO was not produced in high enough concentration in the short residence times to affect the absorption spectra of CH₂OO in 363-395 nm. Only when residence time and reactant concentrations were increased by more than 10 times, weak rovibronic features from the $\tilde{a}^3A_2 \leftarrow \tilde{X}^1A_1$ transition of HCHO can be identified in this wavelength region (see Extended Data Fig. 2). This confirms that residence times smaller than 500 ms were short enough to avoid spectra interference from of other reaction byproducts along the reaction cell.”

Specifically, in the spectral range of 363 – 395 nm where CH₂OO was measured, the HCHO peak cross sections are 4 orders of magnitude lower than those for CH₂OO, and thus, HCHO concentrations lower than $2 \times 10^{14} \text{ cm}^{-3}$ are not detectable in this range. As shown in the HCHO concentrations in Extended Data Fig. 7 (measured in a different spectral region of 328.8 – 330 nm, where the HCHO cross sections are two orders of magnitude higher than in 363-395 nm), the maximum HCHO concentrations at ~ 500 ms, the longest residence time in this study, were $< 7 \times 10^{13} \text{ cm}^{-3}$, well below the HCHO detection limit of $2 \times 10^{14} \text{ cm}^{-3}$ at 363-395 nm.

Line 114: It seems more appropriate to describe the estimates of rate constants as measurements.

Response: Revision has been made to line 128, “Therefore, the rate constants of their reactions with CH₂OO can be measured with some degree of precision.”

Line 117: Please provide a table in the extended data/supplementary information giving initial conditions for all experiments performed. The pressures at which experiments were performed should also be clearly stated in the manuscript.

Response: Table of initial conditions for experiments has been added to the Source Data file, and clarifications on pressure (10 Torr) were added to line 93 and line 132.

Lines 120-125 and 137-142: There is some repetition here. It seems unnecessary to explain how the balance between production and consumption leads to the observed profiles.

Response: The repeated discussion in lines 155-159 (lines 137-142 in the previous version of the manuscript) has been removed.

Line 122 (and elsewhere): The unimolecular decomposition of stabilized CH₂OO has been

shown to be $\ll 1 \text{ s}^{-1}$ under the conditions employed in the current work. It is unlikely the unimolecular decomposition of the sCI is playing any significant role.

Response: Indeed, the unimolecular decomposition of stabilized CH_2OO is insignificant under the conditions employed in the current work. This is shown in the mechanism outlined in Extended Data Table 2 and supplementary Table S1. For completeness, we suggest keeping the unimolecular decomposition in the text, but pointing out its less importance. Relevant discussion has been revised to reflect the small contribution from unimolecular decomposition of stabilized CH_2OO in lines 136-138 and elsewhere, “Subsequently, the interplay of bimolecular reactions and (to a lesser extent) unimolecular decomposition of sCIs becomes evident in consuming CH_2OO ...”

Line 130-135: The more common way to model the observations would be to simply start with the initial concentrations for ethene and ozone and run the model forwards in time. If the flow reactor is under plug flow conditions with negligible axial diffusion (as outlined in the extended data), the simple model should be expected to give similar results? What are the benefits of using the model approach described and how do the results differ from the more simple model?

Response: We appreciate the reviewer’s suggestion regarding the “box model” approach. However, the method described in the manuscript, which utilizes a segmented CSTR model, better aligns with the experimental setup and the steady-state nature of the system. The “box model” approach, which solves the ODE system forward in time starting from initial concentrations, assumes a one-time addition of reactants in the reactor and does not inherently account for the steady-state conditions established by continuous reactant flow and mixing. In contrast, the CSTR model explicitly incorporates the steady addition of reactants and the progressive reaction across the residence time axis.

Furthermore, as detailed in the manuscript, dividing the plug flow reactor into 10 CSTR segments converges to the plug flow reactor profile while allowing for accurate simulation of concentration profiles along the reactor axis. This segmentation is essential because CRDS measures the average concentrations along the flow reactor axis, rather than tracking the time evolution of reactant concentrations. Modeling the concentrations within each segment ensures that the contributions of the individual segments to average concentrations probed by CRDS are correctly represented. However, modeling a single CSTR with the total residence time does not capture the evolution of the concentrations along the flow reactor and, thus, doesn’t provide information for optimizing steady state concentrations.

The plug flow reactor can be effectively modeled as a series of joint CSTRs (refer to *Elements of Chemical Reaction Engineering*¹⁵). This equivalence ensures that the segmented CSTR model retains simplicity while maintaining higher fidelity to the experimental system. Modeling and recording the concentrations of CIs in each CSTR is necessary to obtain their average concentrations along the flow direction. Our current method uses the output concentrations of the previous CSTR segment as the input for the next.

Another alternative method is to model each cross-section slice of the flow reactor as a single CSTR, which means varying reaction time from 0 to the total residence time, then do the average. For example, when using initial concentrations of C_2H_4 and O_3 of 1×10^{17} and $9 \times 10^{14} \text{ cm}^{-3}$ to simulate a residence time of 0.7 s by running the model from 0.07 s to 0.7 s with an interval of 0.07 s, the resultant 10-point average CH_2OO concentration along axis is $6.03 \times 10^{10} \text{ cm}^{-3}$, the same result as from our current CSTR method. If the reviewer is referring to this alternative method, in terms of computational complexity, these two equivalent approaches require a similar amount of calculation.

To clarify, we have added the following explanation to lines 95–99 of the manuscript: “Note that the residence time in the plug flow reactor here represents reaction times spanning from 0 to the nominal total residence time (e.g., the 140 ms residence time represents reaction times from 0 to 140 ms), with the measurement integrating the signal from the point of injection up to the nominal total residence time, probing an average across a horizontal slice in Extended Data Fig. 1.”

Line 142: It would be helpful to plot the percentage contributions to CH₂OO loss as a function of time (perhaps in the extended data or supplementary information).

Response: The plot of percentage contributions to CH₂OO loss as a function of residence time has been added to Supplementary Figure S3. A sentence has been added in lines 169-171, “The plot of percentage contributions to CH₂OO loss as a function of residence time is presented in Supplementary Figure S3.”

Line 147: What is the main source of HCOOH? It is included in the table in the supplementary information as one of several products of CH₂OO + HCHO, in Figure 4 with a different yield, and not in the extended data Table 2 (see also comments below on the rate constants given in the extended data Table 2).

Response: Corresponding contents have been revised in line 166, Figure 4 and Extended Data Table 2. The primary source of HCOOH in our model is the reaction of CH₂OO + HCHO, while the other pathways leading to HCOOH formation (such as CH₂OO unimolecular isomerization) have negligible contribution to the accumulation of HCOOH. The reaction of CH₂OO + HCHO → HCOOH + HCHO takes up to 58 % among the four reaction pathways of CH₂OO + HCHO in our model (see R22 – R25 in supplementary Table S1), close to the upper limit of the HCOOH yield determined by Luo et al.¹⁶ (~37 – 54 %). Previous versions of Figure 4 and Extended Data Table 2 simplified the displaying contents and have been revised.

Line 172: The paper by Vereecken et al. discussed in the manuscript does not report their latest findings on the reaction between CH₂OO + O₃. While the paper mentioned did predict a rate constant of $1 \times 10^{-12} \text{ cm}^3 \text{ s}^{-1}$, later work by the same group (2015) using a higher level of theory revised their prediction to $4 \times 10^{-13} \text{ cm}^3 \text{ s}^{-1}$. The work by Chang et al. has also been shown to be impacted by secondary chemistry involving CH₂OO + IO. Here, and in several other places in the manuscript (references to spectra by Foreman et al., product yields for CH₂OO + HCHO, results of Copeland et al. for example) the authors are a little too selective in the previous data they choose to discuss.

Response: The latest work by Vereecken et al.¹⁷ has been cited in lines 192-193, “Theoretical work from Vereecken et al. obtained a rate constant of $4 \times 10^{-13} \text{ cm}^3 \text{ s}^{-1}$ ”. In addition, as in lines 193-199, other literature rate constants of reaction CH₂OO + O₃ by Onel et al., Copeland et al., and Chang et al. were all discussed in comparison with our value in this study.

The other issues about reference CH₂OO spectra, and product yields for CH₂OO + HCHO have been resolved according to the other comments.

Lines 174, 178, 184 (and elsewhere): Where results of previous work are given, uncertainties should be included throughout.

Response: The uncertainties of the cited works have been added to lines 178–199.

Line 184: Luo et al. also reported product yields for CH₂OO + HCHO at similar pressures to those used in the current work. Are these values used in the model? HCOOH is an important potential product of the reaction, and reacts rapidly with CH₂OO.

Response: The reaction of $\text{CH}_2\text{OO} + \text{HCHO} \rightarrow \text{HCOOH} + \text{HCHO}$ takes up to 58 % among the four reaction pathways of $\text{CH}_2\text{OO} + \text{HCHO}$ from our simulation results (see R22 – R25 in supplementary Table S1), close to the upper limit of the HCOOH yield determined by Luo et al.¹⁶ (~37 – 54 %). The reference from Luo et al. was already mentioned in lines 204-205.

Line 198: Can the results of the current work be used to say anything about expected SCI yields at atmospheric pressure?

Response: The agreement of the current result compared to those from SO_2 scavenging methods^{18,19} helps further validate the indirect measurement methods, and thus, may support the results of sCI yields measured with SO_2 consumption (or H_2SO_4 accumulation) at atmospheric pressure (0.42 ± 0.1 from IUPAC). However, direct extrapolation from 20 Torr to atmospheric pressure is challenging. To extend our method for atmospheric-pressure measurements, maintaining the short residence time (< 0.5 s) at higher pressure has not been achieved yet, which may require upgrading the pumping speed and total flow rates by 10-100 times.

Line 205: The yield of HCHO is usually given as 1, owing to its co-production in the initial ozonolysis reaction. The Copeland et al. study features heavily in the manuscript, but there is an extensive body of work in this area that has been evaluated by IUPAC and others. How do these compare?

Response: HCHO yield in ethene ozonolysis was reported to be 0.6 – 1 in previous studies²⁰⁻²⁶, including works by Herron and Huie 1977 (1)²⁰, Su et al. 1980 (0.63 ± 0.11)²¹, Niki et al. 1981 (0.87)²², Horie and Moortgat 1991 (0.66 ± 0.11)²³, Grosjean and Grosjean 1996 (0.99 ± 0.06)²⁴, Neeb et al. 1998 (0.92)²⁵, Copeland et al. (0.89 ± 0.09)²⁶. It was not clear whether these reported HCHO yields lower than 1 were real or due to experimental uncertainty until 2018, when Pfeifle et al.³ reported that the formation of ketohydroperoxide (KHP) was a minor pathway (~10%) other than the POZ routes (~90%) in their high-level trajectory calculations on ethene ozonolysis. The KHP mechanisms proposed by Pfeifle et al.³ have been supported by recent experimental studies²⁷⁻²⁹ such as Lewin et al.²⁹ The IUPAC database³⁰ on ethene ozonolysis was last updated in 2020, at which time the ~10% KHP pathways was still not widely accepted. But now when we look back, both the recent studies on the KHP pathways,^{3,27-29} and the previous studies that reported HCHO yield of 0.6 – 1²⁰⁻²⁶, support our observation of a HCHO yield of ~88% in this work. The additional references were added to line 228.

Line 213-214: It would be helpful to provide some references to the conjecture mentioned.

Response: Citations were added to lines 236–237.

Line 282: A diagram of the experimental setup would be helpful.

Response: A diagram of the experimental setup was added as supplementary Figure S1.

Line 298: Please give the effective path length and typical values for the ringdown time of the empty cell.

Response: They were added to lines 333-334: “...effective path length of empty cell is ~4000 m and typical ringdown time is ~13 microseconds...”

Lines 305 and 306: Concentrations on the order of 10^{11} cm^{-3} have been reported for CH_2OO in photolytic experiments. The statement that the concentrations in the current work are orders of magnitude lower than those in previous photolytic experiments is not true.

Response: The statement was revised in lines 344-346: "...lower than or close to what would be generated by photolysis of diiodomethane and reaction with oxygen (10^{11} – 10^{13} cm⁻³)...", with additional references.

Line 319: How significant are the reactions in the model? The mechanism is comprehensive, but do all the reactions listed need to be included? What are the key reactions in the model? There are differences between the reactions, rate coefficients, and product yields between the table in the supplementary information and the extended data (Table 2) which should be clarified and explained.

Response: The reactions listed in the full-mechanism supplementary Table S1 are of different importance, and the key reactions of CH₂OO that impact CH₂OO concentrations directly are listed (thus highlighted) in Figure 4 and Extended Data Table 2.

Extended Data Table 2 has now been revised for better readability. To clarify, all the reactions listed in Extended Data Table 2 are the same as the corresponding ones in supplementary Table S1, but we simplified Extended Data Table 2 previously, such as the four pathways of the CH₂OO + HCHO reaction listed in supplementary Table S1 were combined into one in the previous version of Extended Data Table 2, which might have caused the confusion on the "different" rate coefficients and products between the two tables. Now the revised Extended Data Table 2 has removed this issue.

Line 333: Are concentrations of O₃, C₂H₄, and HCHO in the model constrained to observations or free-running after initialisation? If free-running, how do they compare to observations?

Response: Corresponding part was revised in lines 373-377, "The reaction rates of CH₂OO with ethene, O₃, and HCHO have been directly determined with the photolysis method and these kinetic data were used as starting points for fitting the model. The concentrations of all the species in the model were free running after initialization, while the reaction rate constants were varied between different runs." The concentrations of O₃, C₂H₄, and HCHO in the model were free running after initialization. The concentrations of HCHO from the model are compared with experimental observations in Extended Data Figure 7, showing good agreement. Simulated and experimental [O₃] (or [C₂H₄]) are not plotted for comparisons because the changes in O₃ (and C₂H₄) concentrations were smaller than 6×10^{13} cm⁻³ due to the short residence times (< 0.5 s). If plotted, both would look like "flat" lines. Another way might be to plot Δ [O₃] for comparisons, yet the small Δ [O₃] values cannot be quantified accurately from the large broad features of O₃ in the CRDS experiments, since the small difference derived from the large [O₃] values (with initial [O₃]_i ~ 0.9 – 1.8×10^{15} cm⁻³) tends to have significant uncertainties.

Line 336: Kinetics of unimolecular processes are only estimated by Chhantyal-Pun et al. More detailed studies have been performed which show the unimolecular decomposition of the SCI to be insignificant under the conditions of the current work.

Response: The other unimolecular decomposition studies of stabilized CH₂OO were cited in lines 377-379, "Stabilized CH₂OO can also undergo unimolecular processes and are described in the model based on data from previous studies^{21,56,73-75} and the evaluated kinetic data⁷¹". The unimolecular decomposition of stabilized CH₂OO turns out to be insignificant but is still included in the mechanism for completeness.

Line 533 (and elsewhere): The primary reference for the data should be given in place of (or in addition to) the reference to the MPI-Mainz UV/Vis Spectral atlas.

Response: Citation of the SO₂ reference (Vandaele et al.³¹) was added to caption of Figure 2. Other similar ones were also treated in the text: citation of the SO₂ reference (Vandaele et al.³¹) was also added to caption of Extended Data Figure 3.

Figure 3: What is the difference between the open symbols and the dashed lines? The dashed lines appear to be from the simulations but do not go through all the open symbols. Can uncertainties be estimated for all the experimental data?

Response: The open symbols are from the simulation. The previous dashed lines used “B-spline” curves to connect the open symbols. Now the line type has been changed to “Straight” connections to make sure they go through the center of all the open points. This has been clarified in caption of Figure 3, “The dashed lines connect the kinetic simulation open symbols under the different reaction conditions.”

The uncertainties for all the experimental data of CH₂OO concentration (except the uncertainty of $8 \times 10^9 \text{ cm}^{-3}$ at 174 ms with high alkene and high ozone concentrations) are estimated to be $1 - 3 \times 10^{10} \text{ cm}^{-3}$ and added to Figure 3. A clarification is added in lines 141-145, “The concentration of CH₂OO at 174 ms with relatively high alkene and high ozone concentrations (solid black square) was measured three times to generate the 1σ error bar ($8 \times 10^9 \text{ cm}^{-3}$) at that point, while spectra baseline fluctuations ($\Delta\alpha$) were used to estimate the error bars of other data points ($\sim 1 - 3 \times 10^{10} \text{ cm}^{-3}$) in Fig. 3 (assuming no additional uncertainties from the CH₂OO reference).” Source Data lists the original spectra data baseline noises, [CH₂OO] concentrations and estimated errors.

Extended data Figure 1 (and elsewhere): In some places, units include “molecules” and in other places they do not. Either should be acceptable, but there should be consistency.

Response: The mentioned units have been unified to “cm⁻³” without “molecules” throughout the text.

Extended data Figure 2: The reference spectrum appears to be at a much lower resolution than the observations. A reference spectrum with similar resolution should be used. Effective absorption cross-sections for HCHO are likely impacted by the spectral resolution, and as such the concentrations of HCHO reported throughout may be impacted, with potential consequences for the conclusions regarding the importance of CH₂OO + HCHO. What other species might be expected to be present and to contribute to the broad absorption?

Response: We agree that the reference spectrum used in Extended Data Figure 2 (Bogumil et al.³²) does not have enough resolution to accurately determine the HCHO concentration from comparison with ours, but this is the only HCHO reference that spans to the wavelength region >370 nm (where CH₂OO was measured), to the best of our knowledge. Therefore, it served well the purpose to show that in the 363-395 nm region where CH₂OO was measured, HCHO contributed little or no inference at residence time < 0.5 ms. As discussed in lines 109-116: “HCHO was not produced in high enough concentration in the short residence times to affect the absorption spectra of CH₂OO in 363-395 nm.” Specifically, the detection limit of HCHO in 363-395 nm was estimated to be $\sim 2 \times 10^{14} \text{ cm}^{-3}$ (using the Bogumil et al. cross sections), whereas at residence time < 0.5 ms, the maximum [HCHO] was $< 7 \times 10^{13} \text{ cm}^{-3}$ (in Extended Data Fig. 7, measured in a different spectral region of 328–330 nm, see the following), well below the HCHO detection limit of $2 \times 10^{14} \text{ cm}^{-3}$ at 363-395 nm.

Most studies of HCHO spectra^{14,32-34} focus on <360 nm regions where HCHO rovibronic features are stronger. Consequently, it needs to be clarified that our other HCHO measurements (those actually used for kinetic modeling of HCHO) in Extended Data Figure 7 used a different wavelength range (328 – 330 nm rovibronic features of HCHO which allow higher accuracy)

and a different, high-resolution reference spectrum (Smith et al.¹⁴), as detailed in the Methods section. Therefore, the concentrations of HCHO obtained for the kinetic simulation in Extended Data Figure 7 and the evaluation on the CH₂OO + HCHO reaction were not affected by the resolution of the >370 nm reference spectra by Bogumil et al.³² A clarification has been added to caption of Extended Data Figure 7, “The HCHO concentrations were obtained in the spectral range of 325 – 340 nm using high-resolution reference spectra by Smith et al.⁶².”

Possible contributors to the broad absorption in Extended Data Figure 2 include CH₂OO, the remaining O₃ (initial concentration $\sim 1 \times 10^{16}$ cm⁻³, cross section $0.5 - 1 \times 10^{-24}$ cm²), or secondary products and radicals.

Extended data Figure 5: It’s not clear that there are any differences between the fits. It would be helpful to tabulate the values for the sCI yield and rate constants derived from the fits shown in the different panels. Do the results from the different fits agree?

Response: The fitting parameters are tabulated and compared in the Source Data (sub-sheet named Extended data Figure 5(a-c)). The results are in good agreement.

Extended data Figure 6: Uncertainties should ideally be estimated for all experimental data points. What are the implications of the results for the sCI yield at atmospheric pressure?

Response: Revised uncertainties are added to lines 210-214: “From the fitting of kinetic modeling with the experimental [CH₂OO] data, the yield of stabilized CH₂OO was determined to be 0.25 (± 0.07) in the low-pressure region (~ 10 Torr). The error bar of the sCI yield from kinetic simulation (relative error $\sim 4\%$, see Extended Data Fig. 5) was much smaller than the uncertainty of [CH₂OO] originated from the CH₂OO reference spectra (relative error $\sim 30\%$), with the latter contributing to most of the sCI yield uncertainty”. Revision was also made in the caption of Extended Data Figure 6, “...those of the red squares at each pressure in this work are estimated from the uncertainty in kinetic simulation (presented in Extended Data Fig. 5) and the uncertainty of [CH₂OO] from the CH₂OO reference spectra.”

The agreement of the current result at ~ 10 Torr compared to those from SO₂ scavenging methods^{18,19} near 10 Torr helps validate the previous works, and thus, might indirectly support the results of sCI yields measured with SO₂ consumption (or H₂SO₄ accumulation) at atmospheric pressure (0.42 ± 0.1 from IUPAC).

Extended data Figure 8: The caption to the figure gives the same initial ozone and ethene concentrations for all data, but the observed and modelled data for CH₂OO at [SO₂] = 4×10^{13} cm⁻³ are higher than those at [SO₂] = 2×10^{13} cm⁻³. Please clarify. How do the traces compare at earlier times? It would be better to show the full time series.

Response: The two [SO₂] were flipped, and this typo has been corrected in Extended Data Figure 8.

We agree that the full-time series, as was done for Figure 3, would be beneficial for more accurate fitting. Note that, with SO₂ addition, the CH₂OO signal would rise and peak at earlier times within ~ 5 ms, as suggested by previous kinetic studies³⁵⁻³⁹ on CH₂OO + SO₂, due to rapid consumption of CH₂OO by the SO₂ reaction. However, the shortest residence time that we were able to achieve in our current flow reactor was ~ 10 ms (the signal changes at > 10 ms were small). To obtain shorter residence time close to 1 ms, a large pumping-speed upgrade is required for our current setup, or a significant redesigning of the flow reactor is needed. A discussion has been added on lines 240-244: “One limitation for the kinetic determinations in this work is that the shortest residence time achieved in the current setup was ~ 10 ms, making it challenging to capture the full-time profile for rapid CH₂OO reactions. For instance, in the

CH₂OO + SO₂ reaction, the CH₂OO signal would rise and peak within 5 ms. Achieving a residence time closer to 1 ms would likely require a ~10-fold increase in the pumping speed.”

Extended data Table 2: Are the values reported the results of the fits? The differences with the mechanism given in the supplementary information are not clear. The rate constant for CH₂OO + HCOOH is two orders of magnitude slower than has been reported in the literature – why? Why are no products specified?

Response: Extended Data Table 2 has been revised. A typo in the rate constant for CH₂OO + HCOOH has been corrected.

For clarification, the Extended Data Table 2 lists key reactions of CH₂OO and is part of supplementary Table S1. The rate constant for CH₂OO + HCOOH in the previous Extended Data Table 2 (1.1×10^{-12}) was a typo and now it is corrected to 1.1×10^{-10} (note that this value in the previous supplementary Table S1 was correct and the modeling was done correctly). Extended Data Table 2 has also been revised for better readability, where products are specified for most reactions except unknown dummy products, and the corresponding reaction numbers in supplementary Table S1 are also cross-referenced in Extended Data Table 2.

Reviewer #2 (Remarks to the Author):

This manuscript describes direct in situ measurement of formaldehyde oxide Criegee intermediate (CH_2OO) in the ozonolysis of ethene, with sufficient effective time resolution and fidelity to validate explicit models of the CH_2OO chemistry. That is a substantial achievement that will have a significant impact in understanding the details of this critical prototype ozonolysis experiment. However, there are some aspects of the work that require clarification before publication.

Uncertainties. The demonstration reaction for the new capability is the determination of the CH_2OO stabilization yield in ethene ozonolysis. The reported yields are in reasonable agreement with previous results, but the uncertainty estimates in the present work are really surprisingly small (relative uncertainties of 4%). I think these estimates must not be the absolute uncertainties and need to be explained more clearly. For example, how does uncertainty in the absorption cross section for CH_2OO contribute to the uncertainty in yield? Naïvely I would think it would directly contribute, as the concentration is determined from the absorption. Yet the stated uncertainty in the yield is at least 5 times smaller than that of the reference cross section (the Foreman et al. (ref 31) measurements referenced in the manuscript have a 1σ uncertainty of $\sim 28\%$, according to the table linked from the Mainz database (ref 44)). Moreover, even the uncertainty inherent in the kinetic model seems like it has to be significantly more than 4%. I imagine the actual absolute uncertainty is similar to this group's previous measurements (ref 26).

Response: Revised uncertainties have been added to lines 211-214: “From the fitting of kinetic modeling with the experimental $[\text{CH}_2\text{OO}]$ data, the yield of stabilized CH_2OO was determined to be $0.25 (\pm 0.07)$ in the low-pressure region (~ 10 Torr). The error bar of the sCI yield from kinetic simulation (relative error $\sim 4\%$, see Extended Data Fig. 5) is much smaller than the uncertainty of $[\text{CH}_2\text{OO}]$ originated from the CH_2OO reference spectra (relative error $\sim 30\%$), with the latter contributing to most of the sCI yield uncertainty”. Revision has also been made in the caption of Extended Data Figure 6, “...those of the red squares at each pressure in this work are estimated from the uncertainty in kinetic simulation (presented in Extended Data Fig. 5) and the uncertainty of $[\text{CH}_2\text{OO}]$ from the CH_2OO reference spectra.”

Kinetics. One of the key strengths of this work is the in situ probing of carbonyl oxide in a manner that allows quantitative comparisons to kinetic models. In that regard, a clearer description of the experiment and the constraints on the time response would be valuable. As I gather from schematic figures in previous work (references 42 & 43) and the description provided here, the CRDS probe is along the flow axis.

--- I think that means the measurement (even under the simplified flow assumptions employed) integrates from injection up to the nominal total residence time, so for a given flow rate the measurements is essentially probing an average of a horizontal slice in Extended Data Figure 1. It would be valuable if this were more clearly described. This will guide future researchers in reactor design.

Response: Clarification was added to lines 95–99: “Note that the residence time in the plug flow reactor here represents reaction times spanning from 0 to the nominal total residence time (e.g., the 140 ms residence time represents reaction times from 0 to 140 ms), with the measurement integrating the signal from the point of injection up to the nominal total residence time, probing an average across a horizontal slice in Extended Data Figure 1.” Furthermore, all “Reaction time”s have been changed to “residence time”s throughout the text and figures.

--- The description of the concentration profiles could be clearer, based on what I think the

measurement probes. Some of the wording (“rapidly measure kinetics” (line 73-74) or “complete transient time profiles” (in the abstract line 19, and in line 78)) sounds like a time-resolved measurement rather than a quasi-steady state measurement that probes a tunable range of reaction times. This should be stated more accurately throughout.

Response: We removed or revised the use of terminologies such as “rapidly” and “transient”. On lines 75-76, “...to directly observe and measure kinetics of the transient CH₂OO intermediate in ozonolysis of ethene at short residence times.” In the abstract lines 19-20, “Complete time profiles of CH₂OO produced in ozonolysis under quasi-steady state conditions were observed for the first time.”

--- I think that the demonstration that CH₂OO can be quantitatively probed in an ozonolysis system, in a manner that detailed kinetics can be followed, is the main contribution of the paper, much more important than the specific kinetics measurements. The authors show that suggest that the scavenging of CH₂OO by SO₂ can be quantified (Extended Data Fig. 8). The accuracy of the kinetics is always affected by some of the flow reactor design choices – subsequent work will build on this and improve the kinetics fidelity, if the limitations are clearly described. If the authors could share an analysis of the major limitations on the kinetics determinations, that would improve the impact greatly.

Response: Corresponding discussions were added to lines 240–247: “One limitation for the kinetic determinations in this work is that the shortest residence time achieved in the current setup was ~10 ms, making it challenging to capture the full-time profile for rapid CH₂OO reactions. For instance, in the CH₂OO + SO₂ reaction, the CH₂OO signal would rise and peak within 5 ms^{15,21,56-58}. Achieving a residence time closer to 1 ms would likely require a ~10-fold increase in the pumping speed. Besides, uncertainties in residence time due to flow uniformity or wall losses are expected to be negligible based on the ideal plug flow reactor assessment in Extended Data Table 1, but future studies could further quantify their effects to refine kinetic accuracy.”

Reviewer #3 (Remarks to the Author):

Dear Editor:

This work utilized the high sensitivity of cavity ring down spectroscopy to directly probe the simplest Criegee intermediate CH_2OO in the ozonolysis reaction of C_2H_4 , the most fundamental ozonolysis reaction. To my knowledge, this is the first work of in-situ spectroscopic probe of CH_2OO in the ozonolysis reaction.

Due to the high exothermicity of primary ozonide decomposition, the Criegee intermediates produced in ozonolysis reaction may have high internal energy, very different from the Criegee intermediates produced from photolysis of diiodoalkanes. Thus, it is necessary to study the ozonolysis reaction itself. Direct observation and kinetic measurements of Criegee intermediates in ozonolysis in situ in real time will anchor the reaction mechanisms and greatly improve our fundamental understanding of the whole reaction network.

In this work, the signal-to-noise ratio of the reported cavity ring down spectroscopy of CH_2OO is very nice, not only giving confidence that CH_2OO has been truly probed, but also offering the best version of the UV spectrum of CH_2OO .

The absolute concentration of CH_2OO was measured in the reaction, offering a valuable check of the kinetics of the ozonolysis reaction.

I have a few minor comments as listed below. Further review is not needed.

1) It will be nice if a schematic plot of the flow reactor is given in the SI.

Response: Schematic plot of our experimental setup has been added to supplementary Figure S1.

2) The authors use the term of “reaction time” throughout this manuscript. But after reading it carefully, I think the “reaction time” should be actually “residence time”. For a given residence time, say 140 ms, the reaction time actually spanned from 0 to 140 ms. Thus, the use of “reaction time” is confusing and potentially MISLEADING.

Please clarify this issue when it first appears in the manuscript and use a proper terminology. Related terms like “the transient time profiles of CH_2OO ” should also be defined carefully. There is a similar issue for Figure 3.

Response: Clarification was added to lines 95–99, “Note that the residence time in the plug flow reactor here represents reaction times spanning from 0 to the nominal total residence time (e.g., the 140 ms residence time represents reaction times from 0 to 140 ms), with the measurement integrating the signal from the point of injection up to the nominal total residence time, probing an average across a horizontal slice in Extended Data Fig. 1.” Furthermore, all “Reaction time”s have been changed to “residence time”s throughout the text and figures.

3) Page 3: “The higher spectral resolution in this work (0.01 nm) compared to the reference (~0.12 nm) assures an improved signal-to-noise ratio, allowing more accurate determination of CH_2OO concentration using the vibronic band features spaced by $\sim 600\text{ cm}^{-1}$ with half-peak widths of $\sim 200\text{ cm}^{-1}$ in the following kinetic experiments.”

I do not agree. The higher spectral resolution is different from the improved signal-to-noise ratio. Although both the spectral resolution and the signal-to-noise ratio are greatly improved in this work. They are different aspects. For such broad vibronic bands, a high spectral

resolution will not help. In addition, it is a bit hard to compare “nm” to “cm⁻¹”. Please use the same unit for this paragraph (at least, added in parenthesis).

Response: The discussion has been revised in lines 101–105, and the values in nm have been added: “Both the higher spectral resolution in this work (0.01 nm) compared to the references (~0.12 – 2 nm)^{16-18,38,39} and the improved signal-to-noise ratio allowed more accurate determination of CH₂OO concentration using the vibronic band features spaced by ~8 nm (or 600 cm⁻¹) with half-peak widths of ~3.5 nm (or 200 cm⁻¹) in the following kinetic experiments.”

Specifically, the literature references from Foreman et al.⁹, Ting et al.¹¹ and Mir et al.¹² are added in Figure 1 and supplementary Figure S2. Ting et al.¹¹, Mir et al.¹² and Foreman et al.⁹ have spectral resolution of 2 nm, 2 nm, 1.5 nm and 0.12 nm, respectively, and can resolve or partially resolve the diffuse vibronic bands of CH₂OO. Compared with Foreman et al.⁹, our CRDS spectra have better signal-to-noise ratio (thus higher precision and accuracy) and better resolution (although both can well represent the diffuse vibronic spectral features that have ~8 nm spacings and ~3.5 nm half-peak widths, our spectra have more data points to fit the spectra for better precision and accuracy). Compared with Ting et al.¹¹ and Mir et al.¹², our spectra have similarly good signal-to-noise ratio but much higher spectral resolution that can better represent the diffuse vibronic spectral features (hence, with the similar signal-to-noise ratio, our spectra capture more accurately the peak and valley spectral features and have many more data points to fit the spectra for higher precision and accuracy). Therefore, both the high spectral resolution and the very good signal-to-noise ratio in our spectra contribute to better accuracy of concentration measurements (even for the relatively broad, diffuse vibronic spectra). Because the diffuse vibronic bands of CH₂OO are spaced by approximately 8 nm and have half-peak widths of ~3.5 nm, spectral resolution better than 2-nm, in combination with good signal-to-noise ratio, should be more reliable for quantifying these features.

4) Figure 1 and page 8: “... based on the reference absorption cross sections from Foreman et al.” There are other works that also reported the absolute absorption cross sections of CH₂OO. They must not be ignored, especially those with a higher accuracy. In particular, Foreman et al. stated that “The uncertainty in the number density (~28%) estimate is easily the largest uncertainty in the absorption cross section determination.” This uncertainty is not small.

Response: The other CH₂OO references were added to Figure 1, supplementary Figure S2, and the corresponding discussions in lines 105–109, “The relative uncertainties in CH₂OO absorption cross sections in the references listed (20–30%^{17,38,39}) are larger than our spectra uncertainties (1σ error bar estimated to be ~3–10% from repeated measurements, see Source Data for Fig. 3), therefore the uncertainties of the cross sections determined in this work is ~30%³⁸ and could be improved when more accurate CH₂OO reference become available in the future.”

5) Extended Data Fig. 1: Should “CRDS Segment” in the x-axis be “CSTR Segment” It seems that this figure is an interpolated one (for the CH₂OO concentration between each segments). In my opinion, it is better to show the original results of those 10 CSTR segments without doing any interpolation, although it may look less pretty. If interpolation is really necessary, it should be explained how the interpolation was done.

Response: No interpolation was used in Extended Data Figure 1. Original data are provided in Source Data (sub-table named Extended Data Figure 1). Although only 10 CSTR segments at each residence time, the large number of kinetic simulations at many residence times (at 0.01, 0.1, 1, 10, 20, 40, 50, 60, 70, 105, 120, 140, 170, 180, 190, 220, 250, 290, 350, 400, 430, 600 ms) used in Extended Data Figure 1 was enough to generate such quality of figure, without interpolation. We recommend keeping “CRDS Segment” for the x-axis.

6) Page 7: The authors use “L/min” and “mL/min”, which I believe should be “sL/min” and “smL/min” (the standard mass flow at 273 K and 760 Torr). Please clarify this.

Response: Corresponding units of standard mass flows at 273 K and 760 Torr have been revised on page 8.

7) Please give company names and model numbers (a lot are missing in the current version) of all the important instruments used.

Response: Additional instrument names and model numbers have been added in the Methods section.

8) Others. page 2, please modify “For many decades, ... ” to “For decades, ... ”

page 3, please modify “... to prevent a high production of reaction products along the reaction cell.” to “... to prevent a high production of reaction byproducts along the reaction cell.”

page 5, please modify “as more formaldehyde oxide remains” to “as more CH₂OO remains”.

Response: Revisions were added to line 38, line 115 and line 189.

References in this response letter:

1. Nguyen, T. L., Lee, H., Matthews, D. A., McCarthy, M. C. & Stanton, J. F. Stabilization of the Simplest Criegee Intermediate from the Reaction between Ozone and Ethylene: A High-Level Quantum Chemical and Kinetic Analysis of Ozonolysis. *J. Phys. Chem. A* **119**, 5524-5533 (2015).
2. Yang, L. & Zhang, J. Effect of Carbon Chain Length on Nascent Yields of Stabilized Criegee Intermediates in Ozonolysis of a Series of Terminal Alkenes. *J. Am. Chem. Soc.* **146**, 24591-24601 (2024).
3. Pfeifle, M. *et al.* Nascent Energy Distribution of the Criegee Intermediate CH₂OO from Direct Dynamics Calculations of Primary Ozonide Dissociation. *J. Chem. Phys.* **148**, 174306 (2018).
4. Klippenstein, S. J. Spiers Memorial Lecture: Theory of Unimolecular Reactions. *Faraday Discuss.* **238**, 11-67 (2022).
5. Jasper, A. W., Sivaramakrishnan, R. & Klippenstein, S. J. Nonthermal Rate Constants for CH₄* + X → CH₃ + HX, X = H, O, OH, and O₂. *J. Chem. Phys.* **150** (2019).
6. Burke, M. P., Meng, Q. & Sabaitis, C. Dissociation-Induced Depletion of High-Energy Reactant Molecules as a Mechanism for Pressure-Dependent Rate Constants for Bimolecular Reactions. *Faraday Discuss.* **238**, 355-379 (2022).
7. Plane, J. M. C. & Robertson, S. H. Master Equation Modelling of Non-Equilibrium Chemistry in Stellar Outflows. *Faraday Discuss.* **238**, 461-474 (2022).
8. Womack, C. C., Martin-Drumel, M. A., Brown, G. G., Field, R. W. & McCarthy, M. C. Observation of the Simplest Criegee Intermediate CH₂OO in the Gas-Phase Ozonolysis of Ethylene. *Sci. Adv.* **1**, e1400105 (2015).
9. Foreman, E. S. *et al.* High Resolution Absolute Absorption Cross Sections of the B¹A'-X¹A' Transition of the CH₂OO Biradical. *Phys. Chem. Chem. Phys.* **17**, 32539-32546 (2015).
10. Sheps, L. Absolute Ultraviolet Absorption Spectrum of a Criegee Intermediate CH₂OO. *J. Phys. Chem. Lett.* **4**, 4201-4205 (2013).
11. Ting, W.-L., Chen, Y.-H., Chao, W., Smith, M. C. & Lin, J. J.-M. The UV Absorption Spectrum of the Simplest Criegee Intermediate CH₂OO. *Phys. Chem. Chem. Phys.* **16**, 10438-10443 (2014).
12. Mir, Z. S. *et al.* CH₂OO Criegee intermediate UV absorption cross-sections and kinetics of CH₂OO + CH₂OO and CH₂OO + I as a function of pressure. *Phys. Chem. Chem. Phys.* **22**, 9448-9459 (2020).
13. Beames, J. M., Liu, F., Lu, L. & Lester, M. I. Ultraviolet Spectrum and Photochemistry of the Simplest Criegee Intermediate CH₂OO. *J. Am. Chem. Soc.* **134**, 20045-20048 (2012).
14. Smith, C. A., Pope, F. D., Cronin, B., Parkes, C. B. & Orr-Ewing, A. J. Absorption Cross Sections of Formaldehyde at Wavelengths from 300 to 340 nm at 294 and 245 K. *J. Phys. Chem. A* **110**, 11645-11653 (2006).
15. Fogler, H. S. *Elements of Chemical Reaction Engineering*. 6 edn, (Pearson Education, 2020).
16. Luo, P.-L., Chen, I. Y., Khan, M. A. H. & Shallcross, D. E. Direct Gas-Phase Formation of Formic Acid through Reaction of Criegee Intermediates with Formaldehyde. *Commun. Chem.* **6**, 130 (2023).
17. Vereecken, L., Rickard, A. R., Newland, M. J. & Bloss, W. J. Theoretical study of the reactions of Criegee intermediates with ozone, alkylhydroperoxides, and carbon monoxide. *Phys. Chem. Chem. Phys.* **17**, 23847-23858 (2015).

18. Hatakeyama, S., Kobayashi, H. & Akimoto, H. Gas-Phase Oxidation of SO₂ in the Ozone Olefin Reactions. *J. Phys. Chem.* **88**, 4736-4739 (1984).
19. Yang, L., Campos-Pineda, M. & Zhang, J. Low-Pressure and Nascent Yields of Thermalized Criegee Intermediate in Ozonolysis of Ethene. *J. Phys. Chem. Lett.* **13**, 11496-11502 (2022).
20. Herron, J. T. & Huie, R. E. Stopped-Flow Studies of the Mechanisms of Ozone-Alkene Reactions in the Gas Phase. Ethylene. *J. Am. Chem. Soc.* **99**, 5430-5435 (1977).
21. Su, F., Calvert, J. G. & Shaw, J. H. A FT IR Spectroscopic Study of the Ozone-Ethene Reaction Mechanism in Oxygen-Rich Mixtures. *J. Phys. Chem.* **84**, 239-246 (1980).
22. Niki, H., Maker, P. D., Savage, C. M. & Breitenbach, L. P. A FT-IR Study of a Transitory Product in the Gas-Phase Ozone-Ethylene Reaction. *J. Phys. Chem.* **85**, 1024-1027 (1981).
23. Horie, O. & Moortgat, G. K. Decomposition Pathways of the Excited Criegee Intermediates in the Ozonolysis of Simple Alkenes. *Atmos. Environ.* **25A**, 1881-1896 (1991).
24. Grosjean, E., de Andrade, J. B. & Grosjean, D. Carbonyl Products of the Gas-Phase Reaction of Ozone with Simple Alkenes. *Environ. Sci. Technol.* **30**, 975-983 (1996).
25. Neeb, P., Horie, O. & Moortgat, G. K. The Ethene–Ozone Reaction in the Gas Phase. *J. Phys. Chem. A* **102**, 6778-6785 (1998).
26. Copeland, G., Ghosh, M. V., Shallcross, D. E., Percival, C. J. & Dyke, J. M. A Study of the Ethene-Ozone Reaction with Photoelectron Spectroscopy: Measurement of Product Branching Ratios and Atmospheric Implications. *Phys. Chem. Chem. Phys.* **13**, 14839 (2011).
27. Rousso, A. C., Hansen, N., Jasper, A. W. & Ju, Y. Low-Temperature Oxidation of Ethylene by Ozone in a Jet-Stirred Reactor. *J. Phys. Chem. A* **122**, 8674-8685 (2018).
28. Genossar, N., Porterfield, J. P. & Baraban, J. H. Decomposition of the Simplest Ketohydroperoxide in the Ozonolysis of Ethylene. *Phys. Chem. Chem. Phys.* **22**, 16949-16955 (2020).
29. Lewin, C. S. *et al.* Experimental Evidence for the Elusive Ketohydroperoxide Pathway and the Formation of Glyoxal in Ethylene Ozonolysis. *Chem. Commun.* **58**, 13139-13142 (2022).
30. Cox, R. A. *et al.* Evaluated Kinetic and Photochemical Data for Atmospheric Chemistry: Volume VII – Criegee intermediates. *Atmos. Chem. Phys.* **20**, 13497-13519 (2020).
31. Vandaele, A. C., Hermans, C. & Fally, S. Fourier Transform Measurements of SO₂ Absorption Cross Sections: II. Temperature Dependence in the 29000–44000cm⁻¹ (227–345nm) Region. *J. Quant. Spectrosc. Radiat. Transf.* **110**, 2115-2126 (2009).
32. Bogumil, K. *et al.* Measurements of Molecular Absorption Spectra with the SCIAMACHY Pre-Flight Model: Instrument Characterization and Reference Data for Atmospheric Remote-Sensing in the 230–2380 nm Region. *J. Photochem. Photobiol., A* **157**, 167-184 (2003).
33. Cantrell, C. A., Davidson, J. A., McDaniel, A. H., Shetter, R. E. & Calvert, J. G. Temperature-Dependent Formaldehyde Cross Sections in the Near-Ultraviolet Spectral Region. *J. Phys. Chem.* **94**, 3902-3908 (1990).
34. Meller, R. & Moortgat, G. K. Temperature dependence of the absorption cross sections of formaldehyde between 223 and 323 K in the wavelength range 225–375 nm. *J. Geophys. Res. Atmos.* **105**, 7089-7101 (2000).
35. Welz, O. *et al.* Direct Kinetic Measurements of Criegee Intermediate (CH₂OO) Formed by Reaction of CH₂I with O₂. *Science* **335**, 204-207 (2012).

36. Liu, Y., Bayes, K. D. & Sander, S. P. Measuring Rate Constants for Reactions of the Simplest Criegee Intermediate (CH_2OO) by Monitoring the OH Radical. *J. Phys. Chem. A* **118**, 741-747 (2014).
37. Chhantyal-Pun, R., Davey, A., Shallcross, D. E., Percival, C. J. & Orr-Ewing, A. J. A Kinetic Study of the CH_2OO Criegee Intermediate Self-Reaction, Reaction with SO_2 and Unimolecular Reaction using Cavity Ring-Down Spectroscopy. *Phys. Chem. Chem. Phys.* **17**, 3617-3626 (2015).
38. Howes, N. U. M. *et al.* Kinetic Studies of C1 and C2 Criegee Intermediates with SO_2 Using Laser Flash Photolysis Coupled with Photoionization Mass Spectrometry and Time-Resolved UV Absorption Spectroscopy. *Phys. Chem. Chem. Phys.* **20**, 22218-22227 (2018).
39. Onel, L. *et al.* Kinetics of the Gas Phase Reaction of the Criegee Intermediate CH_2OO with SO_2 as a Function of Temperature. *Phys. Chem. Chem. Phys.* **23**, 19415-19423 (2021).

Dear Editor:

We thank all the reviewers for their additional comments to improve the manuscript and have incorporated the comments and revised the manuscript accordingly.

Point-by-point responses to the reviewers' comments are as follows.

Reviewer #1 (Remarks to the Author):

The authors have clarified most of the points raised in the previous reviews which has improved the manuscript. However, there remain a small number of areas which should be addressed more completely prior to publication.

The previous comment on the timescale for stabilization of Criegee intermediates, likelihood of bimolecular reactions of 'hot' Criegee intermediates, and discrepancies between photolytic measurements and studies of ozonolysis reactions (lines 53-58) has not been satisfactorily addressed. The timescales referred to in the response do not relate to stabilization of the nascent CI, and the studies of bimolecular reactions of 'hot' species refer to very different systems. This should be more explicit in the manuscript, which gives the impression that there is evidence for bimolecular reactions of 'hot' Criegee intermediates in ozonolysis reactions. If such reactions were to occur, faster reactions of Criegee intermediates would be reported in ozonolysis studies compared to photolytic studies – if anything, it is the other way around.

Response: The anticipated stabilization timescale for Criegee intermediates (CIs) produced by ozonolysis is approximately estimated to be on the order of 10 μs (at 20 Torr in this study) or $\ll 1 \mu\text{s}$ (at ambient pressure), based on the collision frequency at 20 Torr and 760 Torr and room temperature and an assumption that stabilization of hot CIs (denoted below as CH_2OO^*) requires hundreds of collisions (considering the average energy transferred in a deactivating collision for CH_2OO , $\langle\Delta E\rangle_{\text{down}}$, of 170 cm^{-1}).¹ These approximate estimates are largely consistent with our previous kinetic simulation and experimental measurements,² and with what are expected in the previous theoretical calculations.^{1,3}

As these rates of stabilization of CH_2OO^* and the competitive or faster rate of decomposition of CH_2OO^* are orders of magnitude faster than the reaction rates of stabilized CIs (sCIs), the steady-state $[\text{CH}_2\text{OO}^*]$ is much lower than $[\text{sCI}]$. Or alternately speaking, the time scale for the presence of $[\text{CH}_2\text{OO}^*]$ is on the order of 10 μs or less, orders of magnitude smaller than the time scale of ms to hundreds of ms of $[\text{sCI}]$ in the kinetics study in this work. Consequently, CH_2OO^* should not contribute measurably to the $[\text{CH}_2\text{OO}]$ measurements in this work. The likelihood of bimolecular reactions of 'hot' CIs in ozonolysis reactions should be very small. Therefore, although bimolecular reactions of CH_2OO^* could, in principle, occur faster than those of sCI, their impact on the overall CH_2OO kinetic measurements in ozonolysis is negligible.

Furthermore, while a few bimolecular reactions of small 'hot' species by recent studies⁴⁻⁷ showed that 'hot' reactions can lead to a much enhanced reaction rate, such studies have not been extended to the bimolecular reactions of hot CIs. There is no direct evidence for bimolecular reactions of 'hot' Criegee intermediates in ozonolysis reactions (or discrepancies between photolytic measurements and studies of ozonolysis reactions) so far, and the statement in the previous version of the manuscript was somewhat speculative.

Therefore, given that this part serves only as introductory background and that the likelihood of bimolecular reactions of ‘hot’ CIs in ozonolysis reactions is expected to be very small, and there is a lack of direct evidence for bimolecular reactions of CH₂OO* in ozonolysis, we have removed the earlier, speculative statement to avoid potential confusion or overinterpretation. We have revised line 57–62 (*note that the line number refers to that in the revised manuscript text with track changes, and the same thereafter*), by removing the speculative discussion, and only kept the short statement on the energy content of the CI product and moved it to line 46–48, “However, due to the highly exothermic POZ decomposition following 1,3 dipolar cycloaddition of ozone to the olefinic bond, the production of Criegee intermediates in ozonolysis is accompanied by high internal energy^{9,26,27}”. We hope that the above information and revisions could be satisfactory.

The responses also still suggest that higher spectral resolution leads to improved signal to noise. This is not the case. The authors are correct that more data points improve the signal to noise, but this is entirely separate from the spectral resolution. An experiment with a greater number of data points, each with lower spectral resolution than the current work, would also improve the signal to noise. The resolution used in the current study does not better capture the peak and valley features in the CH₂OO spectrum as the spacing between them is such that they can be captured with lower resolution measurements.

It is incorrect to state “both the higher spectral resolution in this work (0.01 nm) compared to the references (~0.12-2 nm) and the improved signal-to-noise ratio allowed more accurate determination of CH₂OO concentration...”. Fundamentally, the determination of CH₂OO concentrations in this work is reliant on absorption cross-sections reported in the previous, lower resolution, studies, and it is not possible that the current work can both use those results and report more accurate concentrations.

Response: The treatments on the spectral resolution and the signal-to-noise ratio are now separated, and revisions were made to line 102-103, “The spectral resolution in this work (0.01 nm) is higher compared to those in the references (~0.12 – 2 nm)⁸⁻¹²” and line 105-109, “The good signal-to-noise ratio in this work allowed the determination of CH₂OO concentration using the vibronic band features spaced by ~8 nm (or 600 cm⁻¹) with half-peak widths of ~3.5 nm (or 200 cm⁻¹) in the following kinetic experiments.”

The authors also refer to “uncertainties in of the cross-sections determined in this work”, but no cross-sections have been determined. This requires correction/clarification.

Response: A clarification was added to line 112-113, “... the uncertainties of the cross sections determined in this work by scaling to the reference cross sections from Foreman et al. are 30%.”

In the responses, the authors have clarified the question regarding the yield of HCOOH from CH₂OO + HCHO in the model, but Extended Data Table 2 shows the total rate coefficient for CH₂OO + HCHO and it could be clearer in the manuscript what branching ratios have been used in the model for this reaction.

Response: Revision was added to Extended Data Table 2 on page 31, “*The reaction of CH₂OO + HCHO → HCOOH + HCHO takes up to 58 % among the four reaction pathways of CH₂OO + HCHO in our model (see R22 – R25 in supplementary Table S1).”

In several places, the authors refer to “complete time profiles”, but this is not an entirely

accurate description as there not a return to zero concentrations and elsewhere the authors refer to the challenges capturing “the full time profile”.

Response: In these places, “complete time profiles” have been revised to “time profiles”.

Uncertainties in values reported in previous work have been added in places, but there are still values referred to in the manuscript without uncertainties. This should be corrected throughout.

Response: Revision was added to line 50–52 and 123. Theoretical works on rate constants (Vereecken et al.^{13,14}, Sun et al.¹⁵) and sCI yields^{1,3} did not report uncertainties. Rate constants from Copeland et al.¹⁶ did not report uncertainties.

Reviewer #2 (Remarks to the Author):

I have read the response to my comments and those of the other reviewers and I appreciate the authors' careful consideration of all the suggestions. The revised manuscript is considerably stronger and better highlights the significant results of this work. One minor remaining comment, which I think the authors need not address because it does not affect the end result, is that I do not believe that the uncertainty in the kinetic modeling can be as low as 4% as the authors state. However, I do quite easily accept that other sources of error dominate and that the final error estimates in the manuscript are reasonable.

Reviewer #3 (Remarks to the Author):

Dear Editor:

I feel that the authors have addressed the concerns raised by the reviewers and this work is publishable now. Just a couple of minor suggestions.

1) For the absolute values of CH₂OO cross sections, both JPL and IUPAC are using the results of Ting et al. See [https://uv-vis-spectral-atlas-mainz.org/uvvis/cross_sections/Organics%20\(carbonyls\)/Carbonyl%20oxides/CH₂OO.spc](https://uv-vis-spectral-atlas-mainz.org/uvvis/cross_sections/Organics%20(carbonyls)/Carbonyl%20oxides/CH2OO.spc)

In addition, the error of Ting et al., $(1.23 \pm 0.18) \times 10^{-17} \text{ cm}^2$ at 340 nm, 15%, is smaller than that of Foreman et al. Thus, it is better to use Ting's values for the concentration determination. In addition, lines 105–109, “The relative uncertainties in CH₂OO absorption cross sections in the references listed (20–30%^{17,38,39}) ...” should be revised accordingly.

Response: Revision was made to line 109-110, “The relative uncertainties in CH₂OO absorption cross sections in the references listed (15–30%⁸⁻¹⁰) ...” Ting et al. (15%) has the smallest uncertainty in the CH₂OO absorption cross sections among these references. However, the practical problem using Ting et al. to determine the CH₂OO concentrations is the difficulty in fitting the peak and valley spectral features of the vibronic bands in the spectra (see their mismatches with our spectra and Foreman et al. in Figure 1), especially since we use the 378 – 387 nm feature for Figure 2 and the CH₂OO kinetic measurements in Figure 3.

2) The authors mentioned in their Reply Letter: "Our high-resolution spectra are from one scan without any spectral averaging."

Does this mean that one data point shown in the figure was from the result of a single laser shot?

This should be clarified in the text such that if another group wants to repeat the experiment, they can know how many laser shots are needed.

Further review is not needed.

Response: Revision was added to line 329-330, "The Nd: YAG pumped dye laser typically scanned the near-UV wavelength at 0.01 nm/step with 20 laser shots for data averaging at each step."

References in this response letter:

1. Pfeifle, M. *et al.* Nascent Energy Distribution of the Criegee Intermediate CH₂OO from Direct Dynamics Calculations of Primary Ozonide Dissociation. *J. Chem. Phys.* **148**, 174306 (2018).
2. Yang, L., Campos-Pineda, M. & Zhang, J. Low-Pressure and Nascent Yields of Thermalized Criegee Intermediate in Ozonolysis of Ethene. *J. Phys. Chem. Lett.* **13**, 11496-11502 (2022).
3. Nguyen, T. L., Lee, H., Matthews, D. A., McCarthy, M. C. & Stanton, J. F. Stabilization of the Simplest Criegee Intermediate from the Reaction between Ozone and Ethylene: A High-Level Quantum Chemical and Kinetic Analysis of Ozonolysis. *J. Phys. Chem. A* **119**, 5524-5533 (2015).
4. Jasper, A. W., Sivaramakrishnan, R. & Klippenstein, S. J. Nonthermal Rate Constants for CH₄* + X → CH₃ + HX, X = H, O, OH, and O₂. *J. Chem. Phys.* **150** (2019).
5. Burke, M. P., Meng, Q. & Sabaitis, C. Dissociation-Induced Depletion of High-Energy Reactant Molecules as a Mechanism for Pressure-Dependent Rate Constants for Bimolecular Reactions. *Faraday Discuss.* **238**, 355-379 (2022).
6. Plane, J. M. C. & Robertson, S. H. Master Equation Modelling of Non-Equilibrium Chemistry in Stellar Outflows. *Faraday Discuss.* **238**, 461-474 (2022).
7. Klippenstein, S. J. Spiers Memorial Lecture: Theory of Unimolecular Reactions. *Faraday Discuss.* **238**, 11-67 (2022).
8. Ting, W.-L., Chen, Y.-H., Chao, W., Smith, M. C. & Lin, J. J.-M. The UV Absorption Spectrum of the Simplest Criegee Intermediate CH₂OO. *Phys. Chem. Chem. Phys.* **16**, 10438-10443 (2014).
9. Foreman, E. S. *et al.* High Resolution Absolute Absorption Cross Sections of the B¹A'-X¹A' Transition of the CH₂OO Biradical. *Phys. Chem. Chem. Phys.* **17**, 32539-32546 (2015).
10. Mir, Z. S. *et al.* CH₂OO Criegee intermediate UV absorption cross-sections and kinetics of CH₂OO + CH₂OO and CH₂OO + I as a function of pressure. *Phys. Chem. Chem. Phys.* **22**, 9448-9459 (2020).
11. Beames, J. M., Liu, F., Lu, L. & Lester, M. I. Ultraviolet Spectrum and Photochemistry of the Simplest Criegee Intermediate CH₂OO. *J. Am. Chem. Soc.* **134**, 20045-20048 (2012).

12. Sheps, L. Absolute Ultraviolet Absorption Spectrum of a Criegee Intermediate CH₂OO. *J. Phys. Chem. Lett.* **4**, 4201-4205 (2013).
13. Vereecken, L., Harder, H. & Novelli, A. The Reactions of Criegee Intermediates with Alkenes, Ozone, and Carbonyl Oxides. *Phys. Chem. Chem. Phys.* **16**, 4039 (2014).
14. Vereecken, L., Rickard, A. R., Newland, M. J. & Bloss, W. J. Theoretical study of the reactions of Criegee intermediates with ozone, alkylhydroperoxides, and carbon monoxide. *Phys. Chem. Chem. Phys.* **17**, 23847-23858 (2015).
15. Sun, C., Xu, B., Lv, L. & Zhang, S. Theoretical Investigation on the Reaction Mechanism and Kinetics of a Criegee Intermediate with Ethylene and Acetylene. *Phys. Chem. Chem. Phys.* **21**, 16583-16590 (2019).
16. Copeland, G., Ghosh, M. V., Shallcross, D. E., Percival, C. J. & Dyke, J. M. A Study of the Ethene-Ozone Reaction with Photoelectron Spectroscopy: Measurement of Product Branching Ratios and Atmospheric Implications. *Phys. Chem. Chem. Phys.* **13**, 14839 (2011).